# Development of the Landslide Susceptibility Map of Attica Region, Greece, Based on the Method of Rock Engineering System

Nikolaos Tavoularis [1,*] , George Papathanassiou [2] , Athanassios Ganas [3] and Panagiotis Argyrakis [4]

1   Regional Administration of Attica, Directorate of Technical Works, L. Syggrou St., 80-88,
    117 41 Athens, Greece
2   Department of Civil Engineering, Polytechnic School, Democritus University of Thrace, 671 00 Xanthi, Greece;
    gpapatha@civil.duth.gr
3   Institute of Geodynamics, National Observatory of Athens, 118 10 Athens, Greece; aganas@gein.noa.gr
4   Department of Informatics and Telecommunications, Faculty of Economics and Technology,
    University of Peloponnese, 221 31 Tripolis, Greece; pargyrak@noa.gr
*   Correspondence: ntavoularis@metal.ntua.gr; Tel.: +30-21-3206-5894

**Abstract:** The triggering of slope failures can cause a significant impact on human settlements and infrastructure in cities, coasts, islands and mountains. Therefore, a reliable evaluation of the landslide hazard would help mitigate the effects of such landslides and decrease the relevant risk. The goal of this paper is to develop, for the first time on a regional scale (1:100,000), a landslide susceptibility map for the entire area of the Attica region in Greece. In order to achieve this, a database of slope failures triggered in the Attica Region from 1961 to 2020 was developed and a semi-quantitative heuristic methodology called Rock Engineering System (RES) was applied through an interaction matrix, where ten parameters, selected as controlling factors for the landslide occurrence, were statistically correlated with the spatial distribution of slope failures. The generated model was validated by using historical landslide data, field-verified slope failures and a methodology developed by the Oregon Department of Geology and Mineral Industries, showing a satisfactory correlation between the expected and existing landslide susceptibility level. Having compiled the landslide susceptibility map, studies focusing on landslide risk assessment can be realized in the Attica Region.

**Keywords:** interaction matrix; heuristic; susceptibility; inventory; Greece

## 1. Introduction

Landslide hazard assessment requires a multi-hazard approach, since the types of landslides that will occur usually have different characteristics with different spatial, temporal, and causal factors [1]. The first step towards the evaluation of landslide hazards on a regional scale (e.g., 1:25,000–1:250,000) is the assessment of the relevant susceptibility, which is defined as the likelihood of a landslide occurring in an area in relation to the local geomorphological conditions [2]. In addition, the landslide susceptibility map can be used as an end product in itself [1]. In order to develop a susceptibility map, it is mandatory to first compile an inventory map where the spatial distribution of existing slope failures is shown. It should additionally be pointed out that on a regional scale map is not feasible to discriminate in detail the type of landslide and delineate the runout per failure.

Having developed the landslide inventory map, the likelihood of slope failures i.e., susceptibility, can be assessed by both qualitative and quantitative methods. The former group of methods includes the knowledge-driven methods (direct and indirect mapping), and the latter group includes the data-driven and the physically-based ones [1]. Considering the regional and local scale maps, the knowledge and data-driven approaches are suggested to be applied; for the former approach a geoscientist i.e., geomorphologist, can directly determine the level of susceptibility based on his/her experience and information

related to terrain conditions, while the data-driven mapping statistical models are used in order to forecast likely to landslide areas, based on information obtained from the interrelation between the spatial distribution of landslide conditioning factors and the landslide zones [3]. The most widely applied data-driven approaches are [1]: bivariate statistical analysis, multivariate statistical models and data integration methods like Artificial Neural Network analysis. Bivariate statistical methods (e.g., fuzzy logic, Bayesian combination rules, weights of evidence modelling) are considered as an important tool that can be used in order to analyze which factors play a significant role in slope failure, without taking into account the interdependence of parameters. Multivariate statistical models evaluate the combined relationship between the slope failure and a series of landslide controlling factors. In this type of analysis, all relevant landslide parameters are sampled either on a grid basis or in a slope unit and the presence or absence of landslides is evaluated. These techniques have become standard in regional-scale landslide susceptibility assessment.

Nowadays, the majority of the studies considering landslide susceptibility mapping makes use of digital tools for handling spatial data such as Geographical Information Systems (GIS). Specifically, the GIS-based techniques are considered very suitable for the landslide susceptibility mapping, in which the predisposing factors (e.g., geology, topography) are entered into the GIS environment and combined with the spatial distribution of slope failures i.e., landslide inventory map [3–6]. For the purposes of this study, the semi-quantitative methodology of Rock Engineering System (RES) originally introduced by Hudson [7] was implemented in Greece, particularly in the Attica region for the assessment of landslide susceptibility. This region, which is a county with a size of approximately 3800 km², was selected due to the following reasons:

(i)　　in this region, many cases of slope failures have been reported (Figure 1); the well-known historical landslide of Malakasa (1995) [8] caused serious economic consequences due to the cut-off connection between Athens (the capital city of Greece) and the northern part of Greece; the dangerous, due to rockfalls, segment (located in Kakia Skala) of the National motorway connecting Athens to Patras, some other characteristic rockfall sites such as Alepochori–Psatha, and Alepochori–Schino in Western Attica. Furthermore, rockfalls at particular segments of main streams due to erosion and flash floods, landslides and rockfalls at Attica islands (e.g., Kithira, Salamina, Aegina, Spetses, Hydra, Poros), are some of the most characteristic slope failures that already took place in the administrative region of Attica. Thus, adopting the principle that "*slope failures in the future will be more likely to occur under the conditions which led to past and present instability*" [9], and inventorying and mapping the susceptible to failure slopes provides crucial information for evaluating the future occurrence of landslides in this region.

(ii)　　the existing information considering the landslide occurrences in Attica Region was dispersed in more than one public agency, and was mainly focused on landslides documented along the road network and residential areas, while only a few cases were georeferenced. The slope failures induced at the mountainous areas and at sites that are not directly affecting the manmade environment were either not recorded or probably under-reported. Thus, there is a need for gathering every slope failure that happened till nowadays, for generating reliable hazard maps in order to use them for civil protection actions.

(iii)　　the Attica region concentrates almost half of the Greek population, more than 60% of the industrial production in Greece and high-value properties and infrastructure. For this reason, mapping areas prone to slope failure helps public authorities associated with public works in taking mitigation measures against the increase of risk in potentially dangerous areas, leading to losses of life and investments in such a densely populated county.

(iv)　　the completeness and quality of the available slope failures and thematic geodata.

(v)　　to the author's knowledge, this is the first time that a landslide susceptibility analysis has been conducted on a regional scale (1:100,000), for the whole territory of the Attica

Region. Furthermore, the generated landslide susceptibility map will serve for many authorities related to public works, as a dynamic map for the planning, design, and implementation of a long-term landslide reduction strategy as well as identifying the areas where more detailed investigations will be required for the planning of critical infrastructure.

(vi)  taking into account that the next five to ten years, very important civil engineering projects are about to be constructed in Attica county (such as transports network elements: highways, railroads, metro-tunnels, hospitals, administrative buildings, security/emergency structures, residential buildings) the existence of a regional-scale landslide susceptibility map could be a very useful tool for supporting decisions in order to prevent the location of high-value constructions in unsuitable locations.

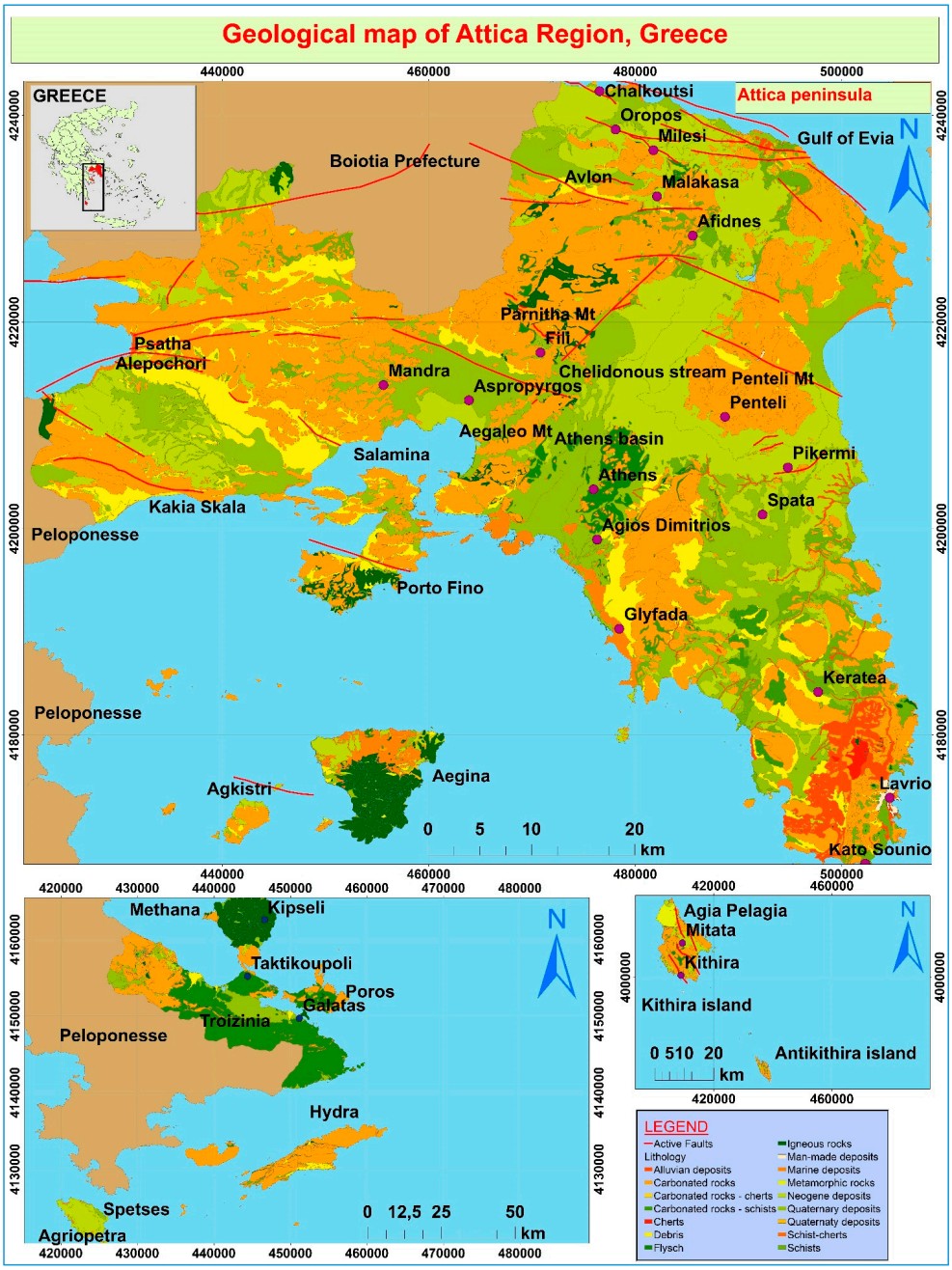

**Figure 1.** Simplified geological map of Attica region, based on the official Greek projection system (EGSA 87). Active faults were inserted in this map from the National Observatory of Athens (NOAFAULTs, https://zenodo.org/record/4304613# .YAmJbugza1Z).

Regarding the above-mentioned, the scope of this study that is part of the project "Landslide Risk Assessment of Attica Region (DIAS)", is twofold: (i) construct a uniform and updated geodatabase of slope failures induced the last sixty years in the whole territory of Attica Region, and (ii) compile a landslide susceptibility map, being the basic step to produce the upcoming landslide hazard and risk maps.

## 2. Geology and Tectonic Setting of Attica

Attica is located in the back-arc area of the Hellenic Arc. The geology of Attica comprises Alpine basement rocks, both metamorphic and non-metamorphic, and post-Alpine sediments (Figure 1). The Alpine rocks belong to the high-pressure metamorphic units of the Cyclades and Almyropotamos that extend from Penteli Mt, east Attica [10] to the southern Gulf of Evia and to the non-metamorphic units of Eastern Greece/Sub-Pelagonian units that outcrop in Parnitha Mt and in west Attica. The southern parts of Attica are also underlain by schists and marbles of the Cycladic Metamorphic Belt. An 8.2 Ma granodiorite outcrops in the Lavrion area of SE Attica. The post-alpine (syn-rift) formations consist of alternating beds of marls, lacustrine limestone marls and sandstones. Quaternary deposits are talus cones, sandy–clayey soils, scree, and unconsolidated clays. [11].

Rifting started in Middle-Upper Miocene and continues until the present day resulting in the formation of several basins. According to Freyberg [12], in the western part of the Athens Basin, the Pliocene formation (with a considerable thickness reaching locally more than 300–400 m) can be found, such as clays, sands and sandstones, and gravels in alternation with white limestone. The dating of the synrift ranges from Upper Miocene to Holocene times. There are also Quaternary volcanic formations consisting of loose volcanic extrusive rocks with tuff blocks, dacitic and andesite domes as well as alluvial fan deposits and steep talus cones covering parts of Aegina island, Poros island and almost the entire Methana peninsula.

The Athens basin is the main neotectonic feature in Attica, elongated in a NE-SW direction. An important tectonic structure is the NNE-SSW, west-dipping detachment fault that separates the metamorphic units to the east from the un-metamorphic units to the west [13,14]. The fault was active in Late Miocene-Early Pliocene and produced several hundred meters of debris-flow deposits. In addition, the active normal faults of Avlon-Malakasa, Afidnes, Milesi, Pendeli, Kakia Skala, Thriassion and Fili dominate the area [15,16]. These faults present characteristic features such as prominent scarp linearity, considerable scarp height, unweathered scarp appearance and fault-slip kinematics that are compatible with the regional stress–strain fields (N-S to NNE-SSW) [17,18].

Based on their morphotectonic features [16], all normal NW-SE trending major faults of Attica could be considered "active structures". Overall, the northern part of Attica is bounded by a series of north-dipping active fault segments, while the central part by south-dipping active faults, respectively [16,19–22]. The slip rates of active faults are less than 1 mm/year [15,21,22] and average earthquake recurrence intervals are expected in the order of a few thousands years.

An interesting part of the geological setting of the Attica region is Kithira and Antikithira islands, which are the southeastern islands of the Ionian Sea between Peloponnese and Crete and belong to the administration of Attica Regional Authority. The geological formations that are found there, comprise metamorphic rocks as well as carbonate rocks of Tripolis and Pindos geotectonic zone. Both islands are surrounded by N-S oriented active faults due to ongoing east-west extension in this area of the Hellenic Arc.

## 3. Materials and Methods

### 3.1. Landslide Inventory of Attica

The first step towards the compilation of a landslide susceptibility map is the development of a landslide inventory [23]. In this study, the generated inventory map, and the landslide geodatabase, cover a chronological period from 1961 up to the present.

The methods that were used for the generation of the inventory are classified into the following approaches:

- An in-depth collection and review of technical reports (analog and electronic copies) from public authorities, research institutes and newspaper articles
- Field surveys and validation of previously mapped landslides by the authors of past reports
- Airborne and satellite image analysis and interpretation using (a) multi-temporal optical images from Google Earth Pro, (b) processed hillshade imagery extracted from a high-resolution Digital Elevation Model (pixel size of 5 m). we used the 5-m Digital Elevation Model for mapping older landslide features and identify new potential ones. Those landslide areas were delineated based on the guidelines recommended by the protocol of Special Paper 42 from the Oregon Department of Geology and Mineral Industries [24]. The identified slope failures were imported in the ArcGIS database, georeferenced, based on the official Greek projection system (EGSA 87), as: (1) spatial data (mapped as points, lines and polygons) and (2) tabular (descriptive) data in text or numeric form, stored in rows and columns in a database and linked to spatial data [24]. Characteristic examples of the slope failures that were reported in the Attica Region and employed in the DIAS geodatabase are shown in the following Figure 2.

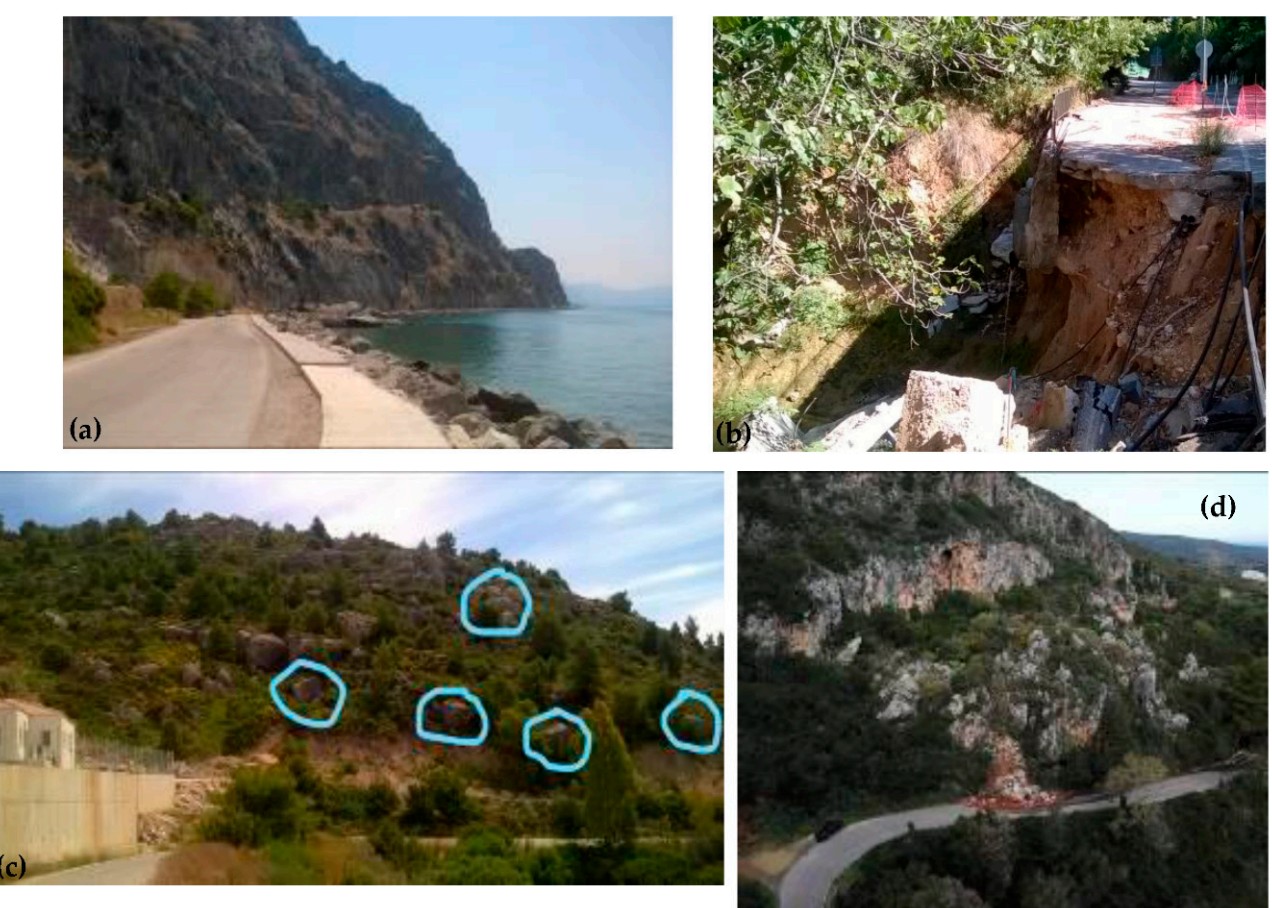

**Figure 2.** *Cont.*

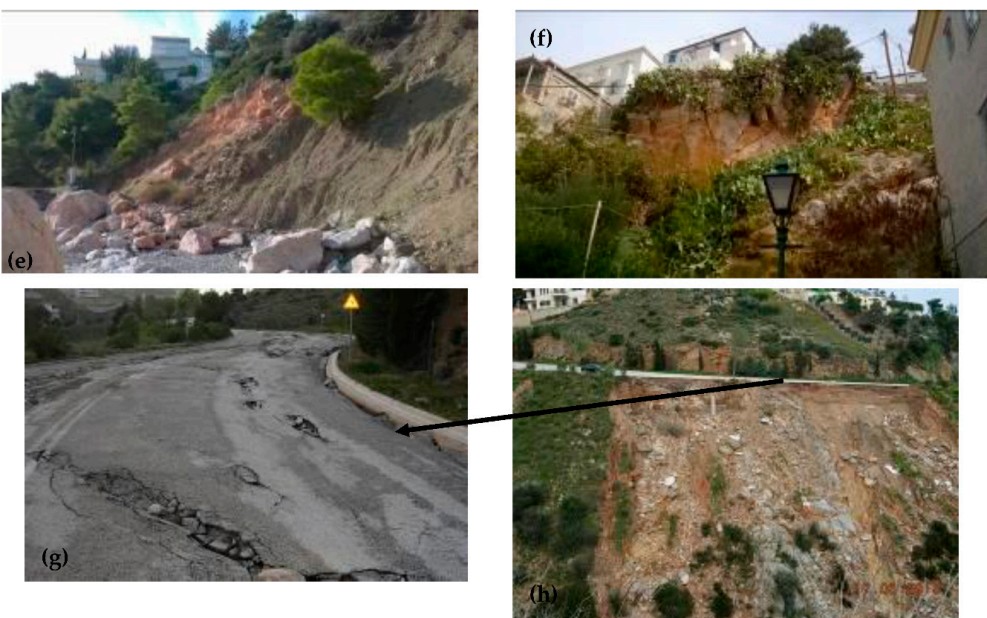

**Figure 2.** (**a**) Rockfalls at the coastal areas of Alepochori–Psatha (North-Western Attica), (**b**) earth fall on bank slopes subjected to undercutting by Chelidonous stream (North of Athens, Kifisia municipality). (**c**) Rockfalls occurred in Attica islands such as Spetses (e.g., Agriopetra) and (**d**) Kithira (e.g., Galani spring-Agia Pelagia). In Figure 2c, the blue circles around rocks emphasize the great possibility for rockfalls. (**e**) Complex slope failure in Salamina island (Porto Fino site), (**f**) A rock topple failure in Hydra (adjacent to Miaoulis statue), (**g**,**h**) An earth slide from Penteli area (Ntrafi site) at northeastern of Athens. The toponyms of each characteristic site are depicted in Figure 1.

Following the terminology defined by the Working Party on World Landslide Inventory (1990) [25], the majority of the depicted slope failure sites hold information on location, dimensions-geometry, landslide-movement type, trigger mechanism, damage caused, slope and aspect, lithological composition, movement date, older activation, seismic risk zone, meteorological data, hydrogeological behavior, consequences, proposed remedial measures, the confidence of landslide identification, mass movement date–field survey date, bibliographic reference and characteristic photos for each slope failure. The developed landslide inventory map is shown in Figure 3, where slope failures are interpreted as points (220 sites), polygons (98 areas delineated based on the Oregon Protocol) and erosion lines based on data provided by the Hellenic Survey of Geology and Mineral Exploration (H.A.G.M.E.), assigning a unique identifier and a number of attributes to each landslide. Taking into account Varnes classification (1978) [26], the movement type of the 220 slope failures, shown as points, can be characterized as follows (Table 1):

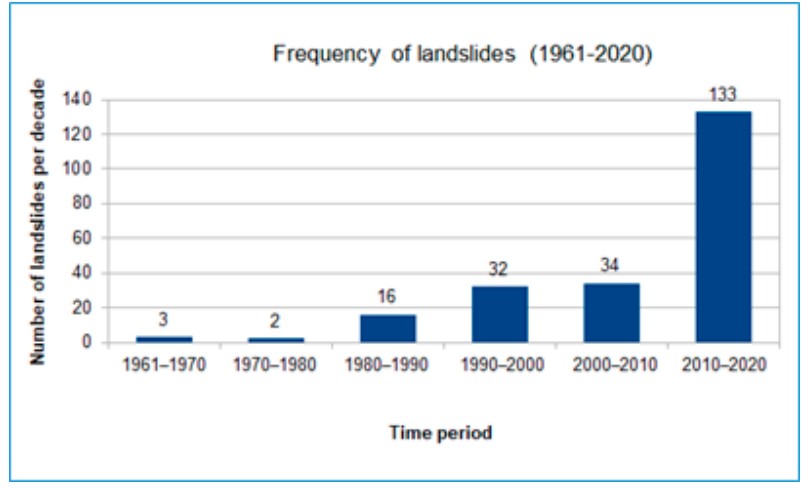

**Figure 3.** Frequency of landslides of Attica for the period 1961–2020. Bin-size ten years.

**Table 1.** Movement type of the 220 inventoried slope failures in Attica Region (based on Varnes nomenclature). Each type is associated with a specific recorded number of failures. Each slope failure is depicted in Figure 4.

| Movement Type | Rock: 52 | Debris: 58 | Earth: 110 |
|---|---|---|---|
| Fall | 1. Rock fall: 40 | 2. Debris fall: 41 | 3. Earth fall: 67 |
| Topple | 4. Rock topple: 3 | 5. Debris topple: 3 | 6. Earth topple: 8 |
| Rotational sliding | 7. Rock slump: - | 8. Debris slump: - | 9. Earth slump: 27 |
| Translational sliding | 10. Block slide: 3 | 11. Debris slide: - | 12. Earth slide: 6 |
| Lateral spreading | 13. Rock spread: - | - | 14. Earth spread: - |
| Flow | 15. Rock creep: - | 16. Talus flow: - | 21. Dry sand flow: - |
| | | 17. Debris flow: 1 | 22. Wet sand flow: - |
| | | 18. Debris avalanche: - | 23. Quick clay flow: - |
| | | 19. Solifluction: - | 24. Earth flow: 1 |
| | | 20. Soil creep: 13 | 25. Rapid earth flow: - |
| | | | 26. Loess flow: 1 |
| Complex | 27. Rock slide-debris avalanche: 6 | 28. Cambering, valley bulging: - | 29. Earth slump-earth flow: - |

The geodata within the DIAS database followed the EU Inspire Directive and is maintained in a digital format that can be adapted and updated for future use. Furthermore, from the DIAS geodatabase, some more extra remarks can be deduced about the frequency of slope failures per decade from 1961–2020 (Figure 3). It is noted that the number of recorded slope failures increased in the 2000–2010 and 2010–2020 decades in comparison to the pre-2000 data, and this can be explained due to intensive climate change and due to the execution of more detailed field and remote sensing surveys from public authorities, research institutes and consulting agencies.

In the following Figure 4, the developed landslide inventory map is shown. In the legend of the map, the slope failures depicted with green circles correspond to landslides that have already manifested at Attica Region in the past. Slope failures in red polygon shapes are those that are delineated through the methodology described by the protocol of Special Paper 42, developed by the Oregon Department of Geology and Mineral Industries. Finally, erosion lines were provided by the Hellenic Survey of Geology and Mineral Exploration (H.S.G.M.E.) through a research project which proposed flooding mitigation measures in the Mandra area, west of Athens.

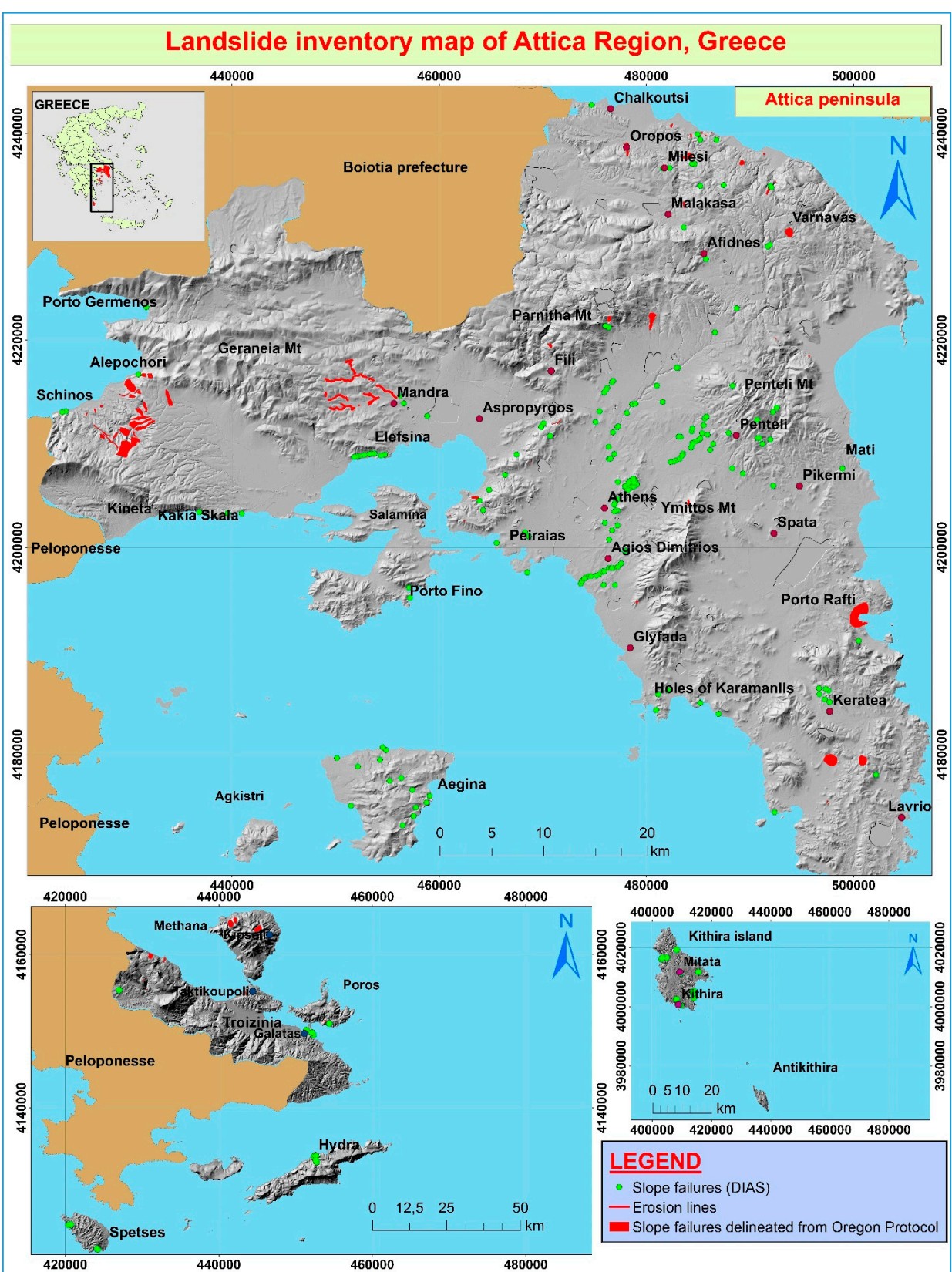

**Figure 4.** The developed by this study landslide inventory map of Attica Region for the period 1961–2020. Background hillshade image is derived from a high-resolution Digital Elevation Model.

*3.2. Assessment of Landslide Susceptibility at the Attica Region*

For the purposes of this study, a semi-quantitative heuristic methodology called Rock Engineering Systems (RES), originally introduced by J. Hudson (1992) [7], was applied to assess the landslide susceptibility. The Rock Engineering System approach has been used for a wide variety of rock engineering and other topics, such as surface blasting, natural slope instability, earthquake and rainfall-induced natural slope instability, road-cut induced slope instability, rockfall assessment, engineering geology zonation, coastal landslides, TBM performance, Metro tunnel stability, and many more applications in engineering modelling and design [27]. Furthermore, regarding recent findings of the implementation of RES generally in geotechnical engineering applications, it can be mentioned that:

(i)     R. Rafiee et al. (2018) [28], have used fuzzy RES in order to apply system thinking-based techniques for assessment of the rock mass cavability in block caving mines.
(ii)    J. Wang et al. (2018) [29], have implemented RES to evaluate sandy soil liquefaction.
(iii)   M. Ferentinou and M. Fakir (2018) [30], used RES in accordance with self-organising maps (e.g., artificial neural networks), so as to assess the stability performance of newly open pit slopes.
(iv)    Finally, M. Elmouttie and P. Dean (2020) [31], used RES and a system theoretic process analysis in order to design the control system for the slope stability monitoring in an open cut mining.

In Greece, the RES methodology has been applied in different geological settings and scales. For example, Rozos et al. (2006) [32] have used RES for a study in Karditsa prefecture, Greece (scaled in 1:50,000), Rozos et al. (2011) [33] have compared RES and Analytical Hierarchy Process (AHP), Tavoularis et al. (2017) [34] tested RES on Malakasa (1995) and Tsakona (2003), Greece in site-specific scale (1:1,000 to 1:5,000), Tavoularis (2017) [35] implemented RES in a regional scale area (Geological Sheet of Megalopolis, Greece scaled in 1:50,000) in complex geological setting and tectonic regime environment.

In this study, an attempt is made to implement RES in a larger coverage area (scaled in 1:100,000) than those previously mentioned with many different geological settings (active faults, places adjacent to dormant volcanic eruptions, streams banks eroded by flash floods), densely populated and surrounded by many important infrastructure facilities.

3.2.1. The RES Approach

A crucial problem of any engineering design is ensuring that all the necessary parameters are included and that the interactions among them are understood. John Hudson was the researcher that originally introduced the Rock Engineering Systems (RES) approach in 1992. The RES methodology is a synthetic approach which studies the problem (e.g., landslide), breaks it down into its constituent variables (e.g., predisposing parameters, estimation of landslide instability index), and assesses their significance (e.g., calculation of susceptibility analysis). In most slopes, that kind of analysis is complicated due to different interacting factors, complexity of geological formations, different scale of the instability events as well as a scarcity of detailed geodata. These problems can be solved through the use of RES, where its use can take into account the particular problems at any investigated site so as to identify critical sites in order to support decisions on land use and planning development [27].

For consideration of a specific engineering project–system (in our research the landslide susceptibility of the Attica region), some parameters are expected to show a greater effect on the project–system than others and some parameters will in their turn be significantly affected by the system. The RES methodology uses a table (i.e., interaction matrix) with $x_i$ rows and $y_j$ columns, in which the selected n parameters are selected as leading diagonal terms and the interactions between them are considered as off-diagonal terms. In Figure 5, the row passing through the parameter Pi represents the influence of Pi on all the other parameters in the system, whereas the column through Pi represents the influence of the other parameters on Pi. Afterward, we study this so-called influence by coding the off-diagonal components in order to express their importance. A semi-quantitative coding

method was used with values ranging from 0 to 4 corresponding to: 0-No interaction (most stable conditions); 1—Weak interaction; 2—Medium interaction; 3—Strong interaction and 4—Critical interaction (most favorable condition for slope failure), respectively. For eliminating the subjectivity, this coding method can be used by one or more experts familiar with the project being considered [7]. Next, the sum of each row (named as "cause-C") and each column (named as "effect-E") can be determined and designated as co-ordinates (C, E) in the diagram of Figure 5. The meaning behind this diagram is that C represents the way in which Pi affects the system; and E represents the effect that the system has on Pi, by indicating a parameter's interaction intensity (as the distance along the diagonal) and dominance (the perpendicular distance from this diagonal to the parameter point). By these two words, we quantify parameter significance inside the matrix system (i.e., landslide).

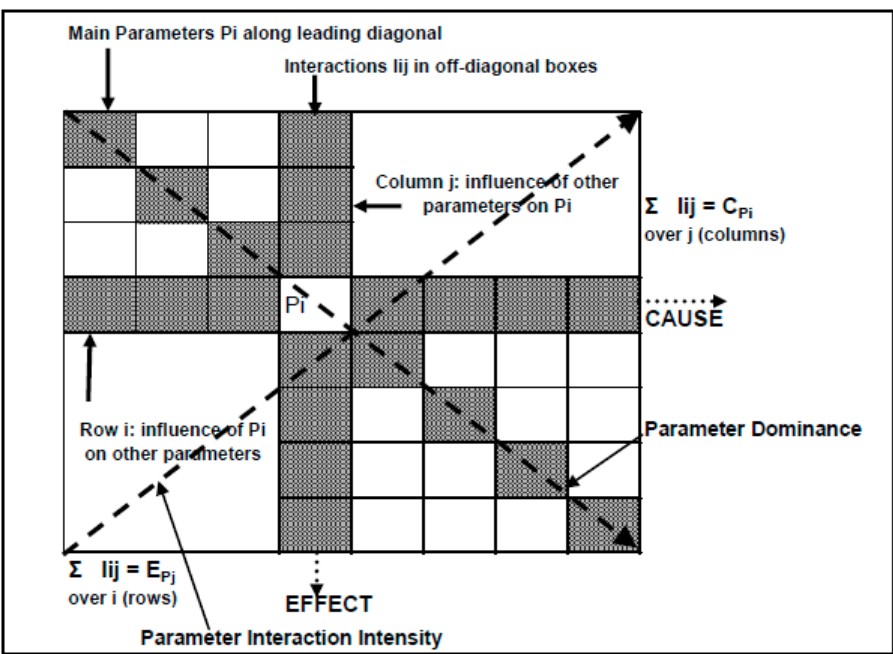

**Figure 5.** Interaction matrix. The dashed lines correspond to the terms "interaction intensity" and "dominance" respectively [7].

According to Hudson (1992) [7], there are many "constellations" that could occur, the two main ones being mainly along the C = E line or mainly along a line perpendicular to it. If the parameter points are scattered along the C = E line but close to it, then they can be ranked according to their parameter interaction intensity; in other words, they can be listed in order of interactive importance (Figure 6a). If, on the other hand, the parameter points are scattered on a line perpendicular to the C = E line, they will have similar interaction intensities but widely differing dominance values (Figure 6b). In the former case, it might be possible to use five or six parameters in such a scheme; in the latter case, all the parameters must be used.

The cause versus effect diagram reveals the influential role of each parameter on slope failure which is expressed by the term "weighted of coefficient influence". Respectively, the role of the system's interactivity is expressed from the histogram of the interactive intensity [cause (C) + effect (E)] against the parameters. This intensity is transformed into weighting coefficients, which express the proportional share of each factor in slope failure and normalized by dividing with the maximum rating (4), giving the ai%, as it is explained in the next paragraph.

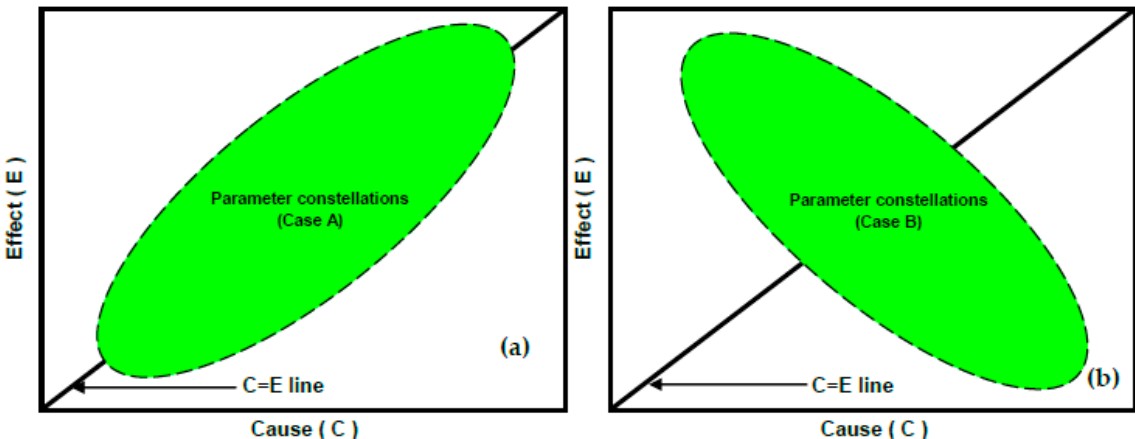

**Figure 6.** Interaction intensity–dominance diagram, with different forms (**a**,**b**) of parameters constellations [34].

The next step is to compute the instability index (Ii) for each examined slope, by using the following equation:

$$Ii = \Sigma a_i \times P_{ij} \tag{1}$$

where $P_{ij}$ is the rating value assigned to the different category of each parameter's separation, i refers to parameters (from 1 to 10 corresponding to this research, and generally from 1 to n, in other case studies where a different number of landslide parameters are selected), j refers to the examined slope and $a_i$ is the weighting coefficient of each parameter provided by the formula:

$$a_i = 1/4 * [(C + E)/(\Sigma iC + \Sigma iE)]\% \tag{2}$$

normalized to the maximum rating of 4. It should be noted that the instability index is an expression of the potential instability of the slope, with values ranging between 0 (no slope failure at all) and 100 which refers to the most unfavorable conditions (i.e., landslide).

### 3.2.2. Selection of the Parameters Controlling the Slope Failures

Ayalew and Yamagishi (2005) have reported that there are five basic concepts for the chosen parameters regarding the assessment of landslide susceptibility [36]. Parameters should: (i) vary spatially, (ii) be measurable, (iii) be related to the presence or absence of landslides, (iv) be representative of the entire study area, and (v) not account for double consequences in the final outcome. Ten parameters were selected as independent controlling factors for the landslide manifestation of the Attica Region, and classified into five classes. These factors which were utilized for the RES methodology are the (i) distance from roads, (ii) slope inclination, (iii) slope orientation (aspect), (iv) lithology, (v) hydrogeological conditions, (vi) rainfall, (vii) land use, (viii) distance from streams, (ix) distance from tectonic elements and (x) elevation.

In order to decide and consequently select the above-mentioned parameters, (a) we studied a huge amount of published and unpublished engineering geology reports, (b) we applied very interesting landslide research based on statistical analysis gained in Greek territory [37], and (c) we took into consideration the field observations conducted in Attica region in the frame of this study [38]. In the following paragraphs, the importance of each selected parameter for the initiation of a landslide and an analysis of what do we mean by the terms "dependence" and "independent" are provided.

The meanings of "dependance" variable and "independent" parameter are related to the role each one has inside the whole system we study. The system can be a slope failure or underground stability and support or the selection of the right type of tunnel boring machine or any other geotechnical engineering problem that can be addressed by using this semi-quantitative heuristic methodology of RES.

To be more specific, referring to "dependance" variable is meant the occurrence or not of a slope failure. For example, we study the interaction of ten landslide parameters

and according to RES methodology, we calculate the weighted coefficient of each landslide parameter, estimating the instability index for each examined slope. If the calculated index is over a critical accepted threshold (as the one that we will present in the following section of this paper), this means that the selected parameters are crucial for the slope failure occurrence, and subsequently, measures must be taken in order to minimize their effect on slope instability. Otherwise, if the estimated index is under this critical threshold, then, no landslide is about to happen and immediately we conclude that those parameters that we selected are not crucial for landslide initiation.

On the other side, as "independent variables" are characterized the landslide controlling factors (such as geology, distance from roads, hydrogeological conditions, distance from tectonic elements), and each other is tested on how dominant or how interactive can be with the other selected landslide parameters.

RES studies the interaction of each parameter to the other and vice versa, by quantifying the different importance of these interactions. This is justified because some parameters will have a greater effect on the system (e.g., in our case the landslide susceptibility in Attica county,) than others and some parameters will in their turn be largely affected by the system. Thus, talking for example about the interaction of hydrogeological condition on lithology, it is meant how lithology can be affected by the permeability status that dictates the geological formation that constitutes the examined slope and vice versa how a specific type of rock or soil of the examined slope will affect the hydrogeological equilibrium of the slope. In another case, we examine how the distance from a road affects the amount of vegetation that exists around this. To be more specific, if a public authority plans to construct a new highway in a place where forest or a grassland area already exists in that particular zone, then it is proved that buffer zones of highway that are in a distance 50 or 200 m from the surrounded slopes affect the existence of vegetation dramatically [33]. Vice versa, the influence of vegetation on slopes that are in an x distance from roads is less important.

In the following paragraphs, a brief comment on the importance of each predisposing landslide parameter is presented.

(i) Distance from roads

During the construction of the road network, vegetation removal, and the application of external loads as well as extensive excavation are some of the most common human intervention actions which are taking place, and result in landslide triggering [39]. It should be mentioned that the digital record of Attica county roads for the generation of DIAS geodatabase was provided by the General Secretary of Civil Protection Agency of Greece. Buffer zones were created around the roads. According to many studies but mostly based on Rozos et al. [33], the slopes that are at a distance of 50 m from a road are more prone to failure.

(ii) Slope inclination

Slope gradient influences on a high grade the slope proneness to failure due to a combination of reasons such as the weathering processes, the internal geometry of geological formations as well as the intensity of meteorological conditions [34]. Through the use of digital elevation model and geographical information systems processing, the slope layer was derived and classified into five classes, as follows: (1) 0°–5°, (2) 5°–15°, (3) 15°–30°, (4) 30°–45°, and (5) >45°, with the higher rating (4) to be given to the slopes with the higher inclination (>45°) [33].

(iii) Slope orientation (aspect)

Another morphological characteristic that influences landslide initiation is the slope orientation (i.e., aspect). Since vegetation and moisture retention depends on aspect, in their turn may affect soil strength and as a result the proneness to landslides. Furthermore, since specific orientations are associated with increased snow concentrations and consequently longer periods for freeze and thaw processes, (not to mention that significant amount of rainfall falling on a slope may vary depending on its orientation [40]),

make everybody accept that this is a very crucial parameter for the estimation of landslide susceptibility. The classification of the slope aspect is shown in Table 2 and its rating is based on Koukis and Ziourkas [37]. According to them, in statistical analysis for landslides in Greece took place in the period 1949–1991, the classes 0°–45°, 45°–90°, are associated more frequently with slope failures. Thus, in this study, the highest rating corresponds to rating 4.

**Table 2.** Parameters and their rating selected to be employed in the model.

| Parameters. | Grade | Parameters | Grade | Parameters | Grade |
|---|---|---|---|---|---|
| 1. Distance from roads | | 5. Hydrogeological conditions | | 9. Distance from tectonic elements | |
| Distant (>200 m) | 0 | Impermeable formations (Marl, siltstone) | 0 | Distant (>200 m) | 0 |
| Moderately distant (151–200 m) | 1 | Fractured formations characterized as having low to negligible permeability (Flysch, schists) | 1 | Moderately distant (151–200 m) | 1 |
| Immediate (101–150 m) | 2 | Volcanic rocks, conglomerate | 2 | Immediate (101–150 m) | 2 |
| Less immediate (51–100 m) | 3 | Carbonate formations with medium to high permeability | 3 | Less immediate (51–100 m) | 3 |
| Close (0–50 m) | 4 | Debris, alluvial–marine deposits | 4 | Close (0–50 m) | 4 |
| 2. Slope's inclination | | 6. Rainfall | | 10. Elevation | |
| 0°–5° | 0 | <400 mm | 1 | >1000 m | 1 |
| 6°–15° | 1 | 400–800 mm | 4 | 0–200 m | 2 |
| 15°–30° | 2 | 800–1000 mm | 3 | 600–1000 m | 3 |
| 30°–45° | 3 | 1000–1400 mm | 2 | 200–600 m | 4 |
| >45° | 4 | 7. Land Use | | | |
| 3. Slope's orientation | | Barren areas | 0 | | |
| 270°–315° | 1 | Urban areas | 1 | | |
| 90°–135°, 135°–180°, 225°–270° | 2 | Forest areas | 2 | | |
| 180°–225°, 315°–0° | 3 | Shrubby areas-Natural grassland | 3 | | |
| 0°–45°, 45°–90° | 4 | Cultivated areas | 4 | | |
| | | 8. Distance from streams | | | |
| 4. Lithology | | Distant (>200 m) | 0 | | |
| Carbonate rocks (e.g., limestones, marbles), schist, cherts | 1 | Moderately distant (151–200 m) | 1 | | |
| Metamorphic rocks exhibiting schistocity | 2 | Immediate (101–150 m) | 2 | | |
| Loose soil formations (alluvial, etc.) | 3 | Less immediate (51–100 m) | 3 | | |
| Flysch, marine deposits | 4 | Close (0–50 m) | 4 | | |

(iv)   Lithology

According to Koukis and Ziourkas [37], lithology in Greek territory is classified into six classes as follows: (a) igneous rocks, (b) cherts, schists, (c) carbonate rocks (e.g., limestones, marbles), (d) metamorphic formations exhibiting schistosity, (e) loose soil formations (alluvial, etc.) and (f) flysch. They concluded that flysch is the geological formation that is associated with the most frequent landslide incidents in Greek territory (36% frequency of landslides), and accordingly it was decided to correspond this complex formation (intercalations mostly of sandstone, siltstone and limestone) to rating 4. In this research, the geologic map of the entire county of Attica region, provided by the Hellenic Survey of Geology and Mineral Exploration (H.S.G.M.E.) was taken into consideration. This map, comprises a digital mosaic of twenty-one (21) geological sheets scaled in 1:50,000.

(v)   Hydrogeological conditions

In this research, the classification is based on River basin management plans from the Greek Ministry of Environment, Energy and Climate Change/Special Secretariat for Water (2012) [41], where the highest rating (4) was given to debris, alluvial–marine deposits whose permeability is crucial for slope failure.

(vi)   Rainfall

It is well known that high precipitation can increase both the groundwater level and the pore pressure in a soil mass/weathered mantle or aquifer, and accordingly it constitutes the main triggering causal factor of landslides [39]. The data that we used were provided by Attica meteorological stations of the National Observatory of Athens (NOA). NOA has published reports presenting the locally encountered conditions [42]. Those data were analyzed using kriging interpolation in order to acquire a rainfall layer of information for the upcoming GIS geoprocessing. In addition, the rating was based on the statistical analysis made by Lalioti and Spanou (2001) for Greece during the period 1991–1998 [43]. In this research, the class 400–800 mm is the one with the greater amount of rain (mean annual) in the Greek territory, so the highest rating for this study corresponds to 4.

(vii)   Land Use

Land use is a crucial parameter in controlling soil erosion as it is related to the vegetation covering which in its turn provides a protective layer on the earth and regulates the transfer of water from the atmosphere to the surface, soil and underlying rocks [44]. The vegetation data used in this study was extracted from the EU Corine Land Cover 2018 database and its rating is based on Rozos et al. [33]. According to them, the higher rating was given to the cultivated areas, due to the maximum percentage of landslide density that is observed.

(viii) Distance from streams

The closer a slope is to a stream, the less stable it is. This happens, due to the fact that streams may adversely affect stability by eroding and saturating the bottom zones of the slopes [45]. The hydrographic network for DIAS geodatabase was generated using the digital elevation model of 5 m pixel size resolution as well as ArcGIS algorithms referring to hydrology processing (Fill, Accumulate, Flow direction based on Strahler classification).

For the examination of this parameter, buffer zones were created around the streams at distances of 50, 100, 150 and 200 m. The classes of the buffer zones are shown in Table 2 and its ranking was based on Rozos et al. [33], suggesting that the most prone class to landslide is that of 0–50 m. This implies that as the distance from the hydrographic axes decreases, the highest percentage of landslide density increases.

(ix)   Distance from tectonic elements (e.g., faults)

There is an increase in the occurrence of slope failures at areas close to fault zones, because as the distance from a tectonic element decreases, the fracture of the rock and the degree of weathering increases [46], while the structure of the surficial material is affected

causing selective erosion and forcing the movement of water along fault planes to decrease slope stability [47,48]. In Attica Region, many active faults were mapped particularly in west and northeastern part of its peninsula as well as in some islands (such as those of Salamina, Kithira). The digital fault database was provided by the Hellenic Survey of Geology and Mineral Exploration and from the National Observatory of Athens [49]. The classes of the buffer zones are shown in Table 2, with the most prone class to landslide to be that of 0–50 m (rating: 4) [33].

(x)  Elevation

The combination of elevation, precipitation and erosion-weathering process contribute to landslide manifestation. The elevation data used in the model were derived from high-resolution DEM (5 m pixel analysis) provided by the Greek Cadastre S.A. The classes of the buffer zones are shown in Table 2 and its ranking was based on the landslide statistical analysis made by Koukis and Ziourkas (1991) for Greece during the period 1949–1991 [37]. In this research, the category 200–600 m is related to the highest number of slope failures that happened in Greece, so this class is associated with a rating of 4.

The above data were rated so as to be used in the development of the interaction matrix (Table 2).

## 4. Results

### 4.1. Implementation of RES for the Estimation of Weighted Coefficients

In this section, the results of the application of the RES method in the Attica Region are presented, such as the interactions among the selected parameters, the calculation of their weighting coefficients and finally the instability index accompanying with charts and tables which they decode and translate the geodata. As it was previously presented, the interaction matrix shown in Table 3 was coded using the Expert Semi-Quantitative method. For example, regarding the effect of lithology (P4) on rainfall (P6), it can be stated that there is no influence at all (coding: 0), whereas rainfall does affect lithology through the infiltrating and weathering-erosion process that may alter not only the mineralogical composition of a specific rock or soil of the slope but also influence their hydrogeological behavior too (coding: 2).

Note that, in Table 3, the sum of cause-and-effect (C + E) value for each parameter represents the "interaction intensity" term, which means how active that parameter is within the matrix system (i.e., the slope stability). On the contrary, the (C − E) value represents how dominant the variable is within the system: positive values of (C − E) represent a dominant variable, whereas negative values of (C − E) represent that the system is affecting the variable more than the variable is affecting the system [7]. More specifically, from Table 3 and Figure 7, it can be seen that the hydrogeological conditions are the most interactive parameter (C + E = 39) [e.g., has the greatest value (concerning C + E)], meaning those conditions play the most decisive role for landslide activation, whereas elevation is the least interactive (C + E = 18). This suggests that elevation does not depend on the influence of the other parameters, but it is an independent agent.

**Table 3.** Coding values for the interaction matrix of Attica Region.

| | | | | | | | | | | | |
|---|---|---|---|---|---|---|---|---|---|---|---|
| **Interaction Matrix of Attica Region** | | | | | | | | | | | |
| P1 | 3 | 1 | 0 | 1 | 0 | 2 | 0 | 0 | 0 | 7 | |
| 2 | P2 | 1 | 0 | 1 | 0 | 2 | 2 | 1 | 0 | 9 | |
| 1 | 2 | P3 | 1 | 2 | 2 | 2 | 2 | 0 | 0 | 12 | |
| 1 | 3 | 2 | P4 | 4 | 0 | 2 | 3 | 2 | 2 | 19 | |
| 2 | 2 | 2 | 2 | P5 | 0 | 3 | 3 | 1 | 0 | 15 | |
| 4 | 3 | 0 | 2 | 4 | P6 | 4 | 3 | 0 | 0 | 20 | Cause (C) |
| 0 | 1 | 0 | 1 | 2 | 0 | P7 | 1 | 0 | 0 | 5 | |
| 2 | 1 | 1 | 1 | 4 | 0 | 2 | P8 | 1 | 0 | 12 | |
| 4 | 3 | 1 | 2 | 4 | 0 | 0 | 2 | P9 | 0 | 16 | |
| 2 | 2 | 0 | 1 | 2 | 4 | 3 | 2 | 0 | P10 | 16 | |
| 18 | 20 | 8 | 10 | 24 | 6 | 20 | 18 | 5 | 2 | | |
| **Effect (E)** | | | | | | | | | | | |

| | | | | |
|---|---|---|---|---|
| P1 = Distance from roads | P2 = Slope | P3 = Aspect | P4 = Lithology | P5 = Hydrogeological conditions |
| P6 = Rainfall | P7 = Land Use | P8 = Distance from streams | P9 = Distance from tectonic elements | P10 = Elevation |

| | Parameters | C + E | C-E | $[(C + E)/\Sigma(C + E)]*100\%$ | Maximum rating | Weighted coefficient $a_i = [(C + E)/\Sigma(C + E)]*100\%/4$ |
|---|---|---|---|---|---|---|
| 1 | Distance from roads | 25 | −11 | 9.54 | 4 | 2.39 |
| 2 | Slope | 29 | −11 | 11.07 | 4 | 2.77 |
| 3 | Aspect | 20 | 4 | 7.63 | 4 | 1.91 |
| 4 | Lithology | 29 | 9 | 11.07 | 4 | 2.77 |
| 5 | Hydrogeological conditions | 39 | −9 | 14.89 | 4 | 3.72 |
| 6 | Rainfall | 26 | 14 | 9.92 | 4 | 2.48 |
| 7 | Land Use | 25 | −15 | 9.54 | 4 | 2.39 |
| 8 | Distance from streams | 30 | −6 | 11.45 | 4 | 2.86 |
| 9 | Distance from tectonic elements | 21 | 11 | 8.02 | 4 | 2.00 |
| 10 | Elevation | 18 | 14 | 6.87 | 4 | 1.72 |
| Total | $\Sigma(C + E)$ | 262 | | 100.00 | | 25.00 |

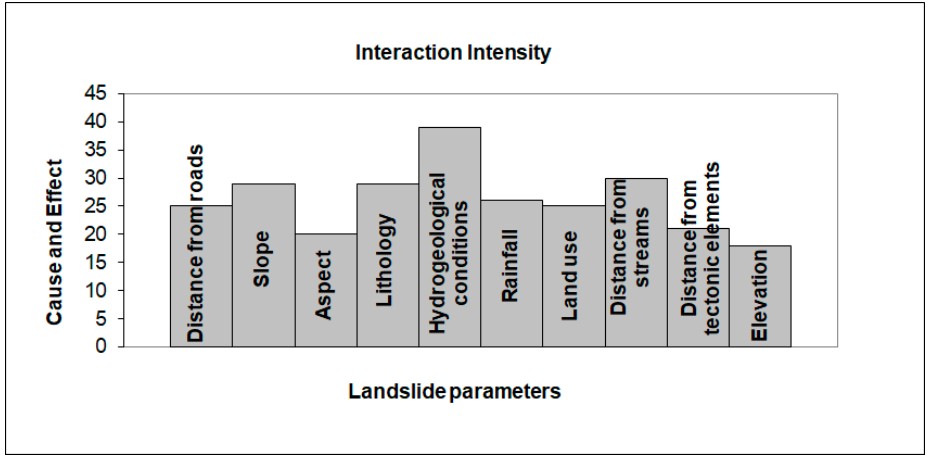

**Figure 7.** Histogram of interaction intensity.

From the RES model and by focusing on the weights assigned to each parameter, it can be clearly reported that hydrogeological conditions contribute the most to landslide occurrence out of all the factors, followed by distance from streams, lithology, slope angle, rainfall, distance from roads, land use, distance to fault lines, aspect and elevation.

In Figure 8, the form of C vs. E constellation in relation to C = E line, defines the number of parameters that will be needed for calculating the instability index. So, according to the interaction intensity–dominance diagram (Figure 6b), the form of the C vs. E constellation is (almost) perpendicular to the C = E line, which means that (based on the aforementioned RES analysis) there is little range in parameter interaction intensity. On the contrary, there is a wide range in dominance (C − E values), so all the selected parameters will be required for the calculation of the instability index for each examined slope.

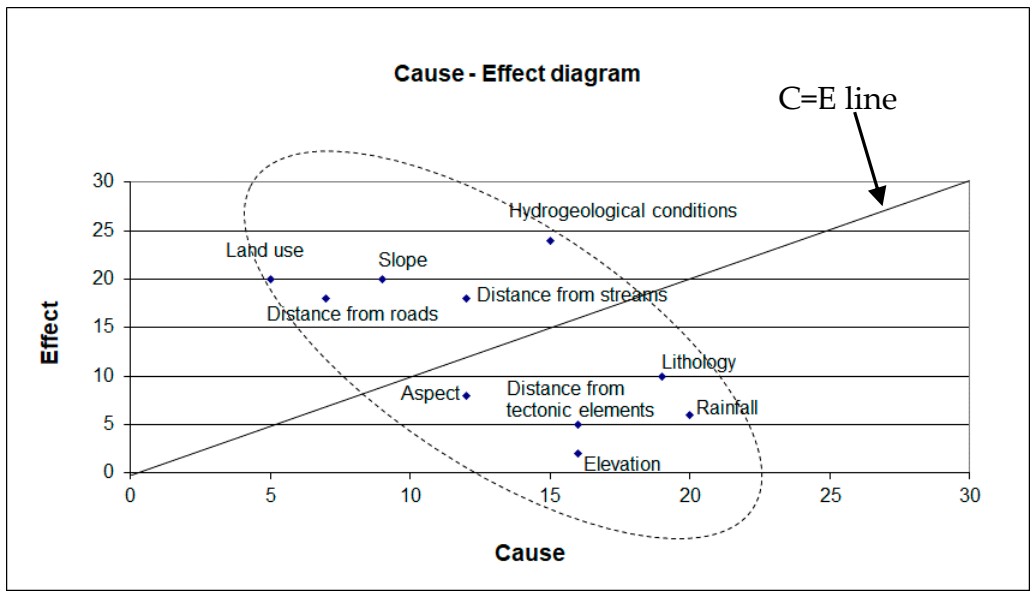

**Figure 8.** Cause–Effect diagram.

Supplementary, the following Table 4, decode and "translate" simultaneously the geodata acquired from our research and contribute in giving the necessary objective answer to the prognosis of the potential instability of the examined slopes of Attica Region. This can be accomplished by the estimation of the instability index, as clearly explained in Section 3.2.1.

A characteristic sample, 10 out of 220 cases of the computation results regarding the instability index, is given in Table 4. In this table, each examined slope (is depicted in the column "Slopes") is ranked according to Table 1 rating, taking into account in parallel the specific geological conditions that characterize it according to either the ad-hoc technical report we collected or field study we carried out. Afterward, for each slope site, every ranking of each parameter (each parameter is depicted in the second line under the title "Parameters", named as 1, 2, 3, . . . , 10) is multiplied by its weighted coefficient (last line of the Table) respectively and each outcome, based on Equation (1) is added in order to yield the instability index for each slope. For example, the instability index of Slope (1) is estimated as follows:

$$\Sigma \ [\text{Parameter (1): } 4 * 2.39 + \text{Parameter (2): } 1 * 2.77 + \ldots + \text{Parameter (10): } 2 * 1.72] = 71 \qquad (3)$$

In Table 5, the classification for relative landslide susceptibility is listed as proposed by Brabb et al. (1972) [50].

**Table 4.** Calculation of Instability Index based on Rock Engineering System methodology for a characteristic sample of 10 slope failures out of 220 ones in Attica Region.

| Slopes/Coordinates (Greek Projection EGSA 87) | Parameters | | | | | | | | | | Instability Index |
| | 1 | 2 | 3 | 4 | 5 | 6 | 7 | 8 | 9 | 10 | |
|---|---|---|---|---|---|---|---|---|---|---|---|
| 1 (476,117–4,215,245) | 4 | 1 | 3 | 4 | 4 | 4 | 1 | 4 | 0 | 2 | 71 |
| 2 (476,790–4,216,087) | 4 | 0 | 3 | 4 | 4 | 4 | 1 | 4 | 0 | 2 | 68 |
| 3 (483,219–4,208,555) | 4 | 1 | 4 | 4 | 1 | 4 | 1 | 4 | 0 | 4 | 65 |
| 4 (482,341–4,208,253) | 4 | 1 | 3 | 4 | 1 | 4 | 1 | 4 | 0 | 4 | 63 |
| 5 (483,441–4,208,871) | 4 | 1 | 2 | 4 | 1 | 4 | 1 | 4 | 0 | 4 | 62 |
| 6 (458,846–4,212,690) | 4 | 1 | 4 | 3 | 1 | 1 | 4 | 4 | 0 | 2 | 59 |
| 7 (477,287–4,211,687) | 4 | 2 | 3 | 4 | 4 | 4 | 1 | 4 | 0 | 2 | 74 |
| 8 (476,938–476,938) | 4 | 1 | 3 | 4 | 4 | 4 | 1 | 4 | 0 | 2 | 71 |
| 9 (475,095–4,212,107) | 4 | 1 | 3 | 4 | 4 | 4 | 1 | 4 | 0 | 2 | 71 |
| 10 (457,187–4,195,149) | 4 | 3 | 2 | 1 | 3 | 1 | 1 | 4 | 4 | 2 | 63 |
| Maximum Pij rating | 4 | 4 | 4 | 4 | 4 | 4 | 4 | 4 | 4 | 4 | |
| [(C + E)/Σ(C + E)] * 100% | 9.54 | 11.07 | 7.63 | 11.07 | 14.89 | 9.92 | 9.54 | 11.45 | 8.02 | 6.87 | 100 |
| Weigh. Coeff. (ai) = C + E)/Σ(C + E)] * 100%/4 | 2.39 | 2.77 | 1.91 | 2.77 | 3.72 | 2.48 | 2.39 | 2.86 | 2.00 | 1.72 | |

**Table 5.** Classification for relative landslide susceptibility proposed by Brabb et al. (1972) [50].

| % Failed Area | 0–1 | 2–8 | 9–25 | 25–42 | 42–53 | 53–70 | 70–100 |
|---|---|---|---|---|---|---|---|
| Relat. Susceptib. | I | II | III | IV | V | VI | L |
| | Negligible | Low | Middle | High | Very high | Extremely high | Landslide |

As it is shown in this table, the generated instability index that is greater than 53%, corresponds to extremely high relative susceptibility up to slope failure and that this is the crucial point for a planner or a researcher for producing a landslide susceptibility map for a particular examined area. This remark is going to be used extensively in the following sessions of this study.

*4.2. Correlation of Spatial Distribution of Slope Failures with the Predisposing Factors Using Statistical Analysis*

Based on the information of Table 4, and according to the ranking of parameters of Table 2, the following useful findings come out during the generation of the susceptibility map of the Attica region. Based on this analysis, it can be concluded that 211 out of 220 (96%) slope failures are in a distance from roads up to 50 m.

Concerning the aspect parameter, 37% of the examined slopes are primarily more abundant on Southeast-facing and secondly on Northwest-facing (34%). Based on the rating assigned to each geological formation (e.g., lithology), the highest (40%) one is observed at flysch (and debris) and secondly to carbonate rocks (37%). This remark was expected since the former ones are the most statistically frequent formations prone to landslides in Greek territory, whereas the latter ones are associated mainly with rockfall incidents in many parts of Greece.

Regarding hydrogeological conditions, carbonate rocks with medium to high permeability due to karstification and secondary fragmentation correspond to the highest (35%) category of permeable rocks in this study. Based on the comparison among rainfall data and landslide occurrences, it was established that landslides are more likely to take place

when the mean annual rainfall is between 400–800 mm. As far as land use parameter is concerned, landslides reported mostly in urban areas (62%) while based on the results given for the elevation, it was found that the landslides develop preferentially on 0–200 m of altitude (63%).

Furthermore, a large portion of landslides (58%) are located near to the hydrographic network in relation especially to the undermining of the banks between 0 m and 50 m. Such places were recorded in many streams (mostly) in the Athens basin (such as those of Kifisos river, Chelidonous, Sapfous, Penteli, Eschatia stream).

Summarizing, the percentages of landslides per each class of predisposing factor are illustrated in the following Figure 9.

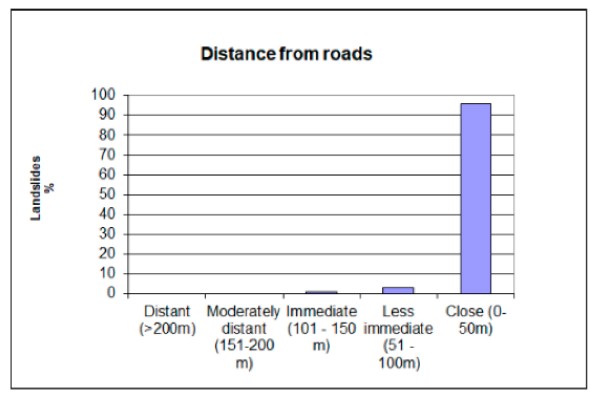
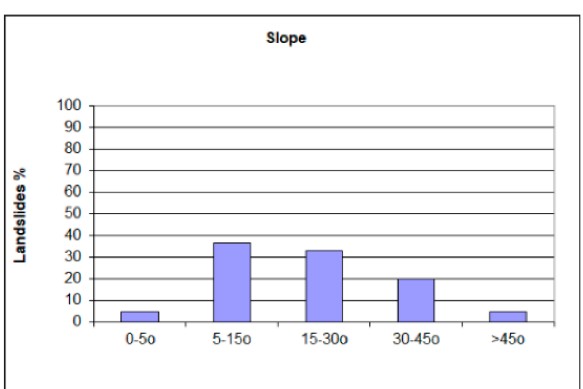

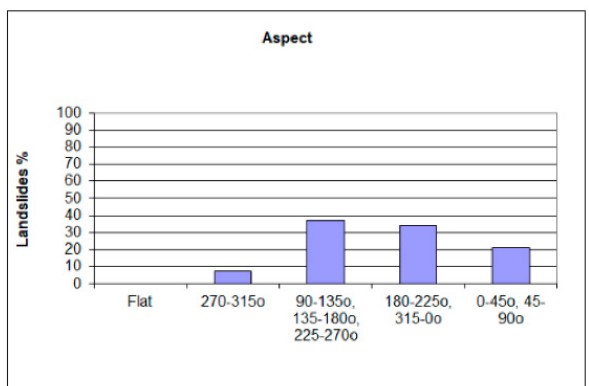
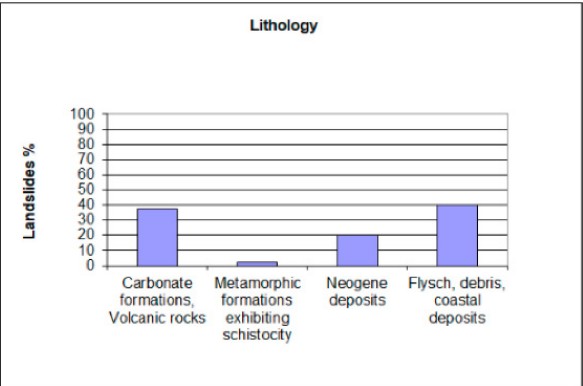

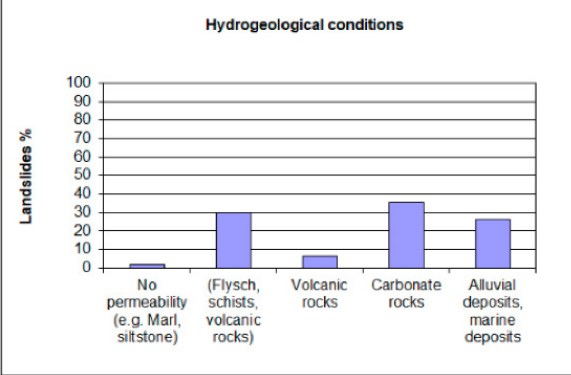
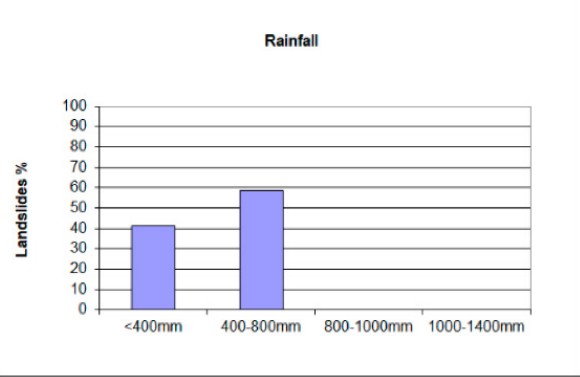

**Figure 9.** *Cont.*

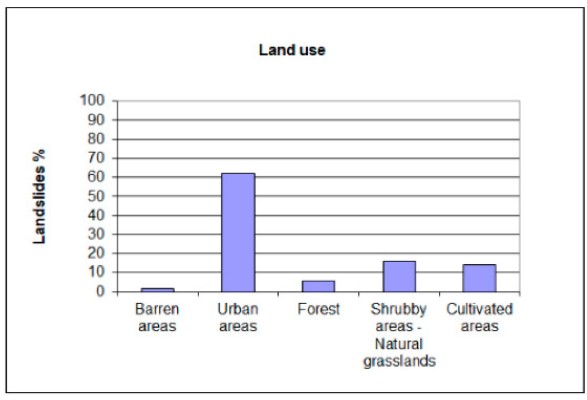
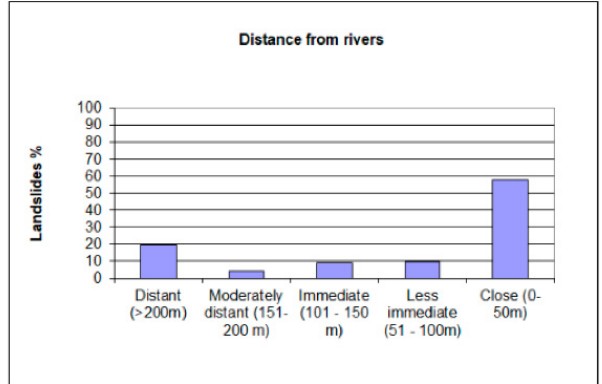
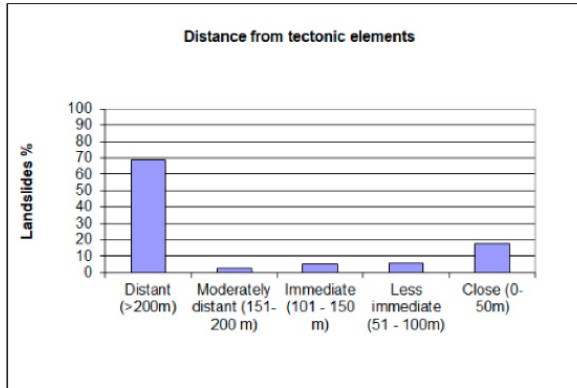
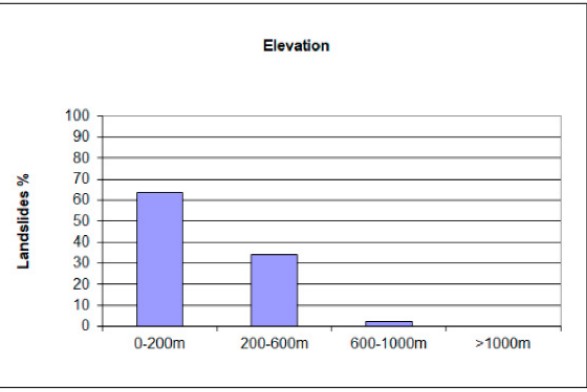

**Figure 9.** Percentage of landslides in each class of the causal and triggering factor of landslide occurrence.

### 4.3. Landslide Susceptibility Map

The subdivision of the predisposing parameters into subclasses (from Table 2) was used for the evaluation of the final slope failure susceptibility map. This map was generated in a GIS environment, through the use of different layers-thematic maps (Figure 10a–j). The data used for the preparation of these layers were obtained from different geodata sources among which are the Digital Elevation Model from Hellenic Cadastre S.A. and a mosaic geological map from the Hellenic Survey of Geological and Mineral Exploration. All data layers were digitized either from the original thematic maps or derived from spatial GIS calculations and finally were converted into grids with a cell size of 20 × 20 m. Afterward, weights and rank values to the reclassified raster layers (representing predisposing factors) and to the classes of each layer were assigned, respectively. This was realized with the use of the previously extended analyzed methodology of RES. Finally, the weighted raster thematic maps with the assigned ranking values for their classes were multiplied by the corresponding weights and added up (through the ArcGIS tool of the weighted sum) to yield the slope failure map where each cell has a certain landslide susceptibility index value. The reclassification of this map represents the final susceptibility map of the study area, divided into susceptibility zones according to Brabb et al. (1972) [50] classification (Figure 11). The landslide susceptibility index (LSI) values in the final susceptibility map were classified into five categories, namely "Low-Middle", with Instability index (Ii) < 25, "High" with 25 < Ii < 42, "Very High" with 42 < Ii < 53, "Extremely high" with 53 < Ii < 70", and "Landslide" with Ii > 70%. From this classification, it can be clearly notified that the higher the LSI, the more susceptible the area is to landslides (instability index higher than 70%).

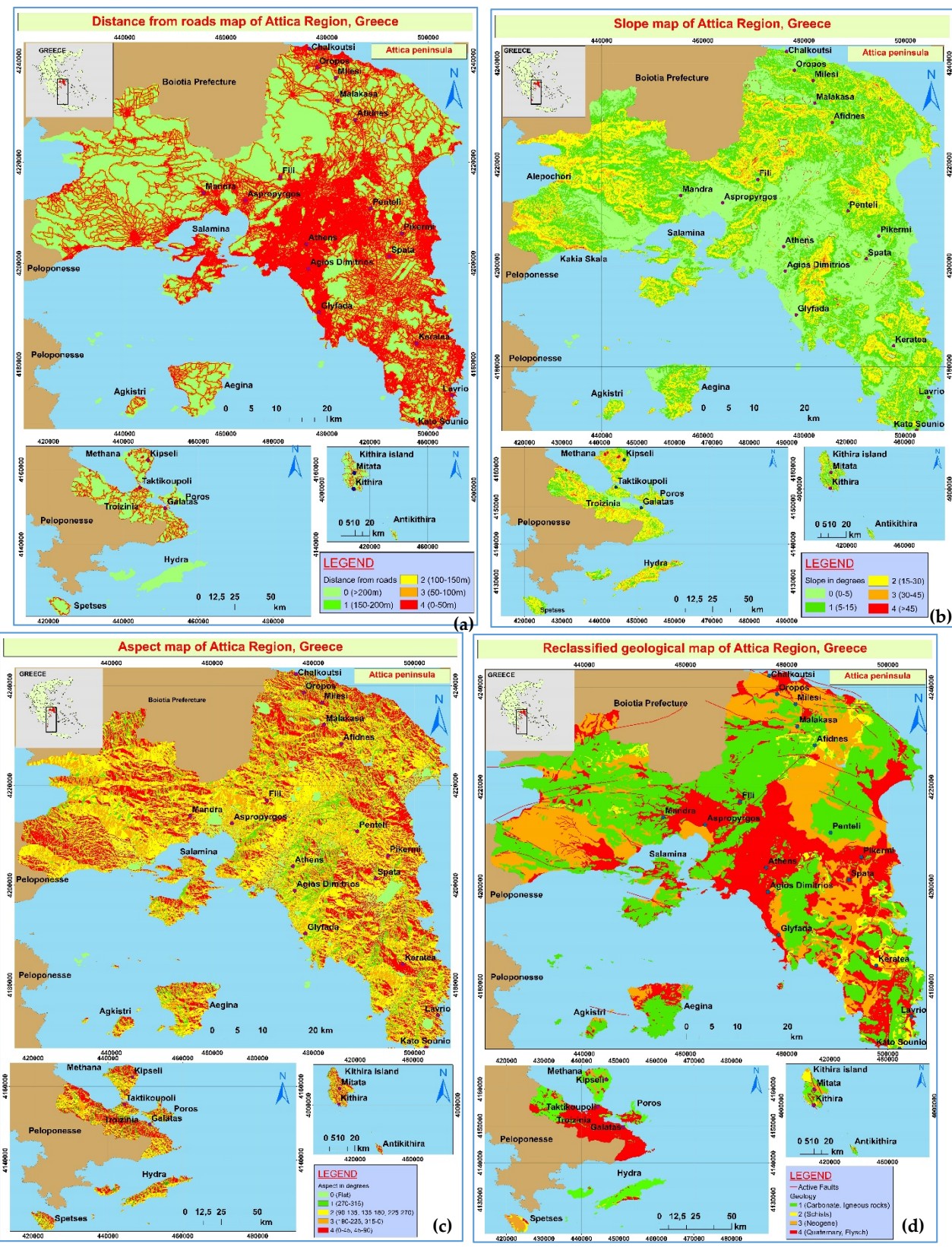

**Figure 10.** *Cont.*

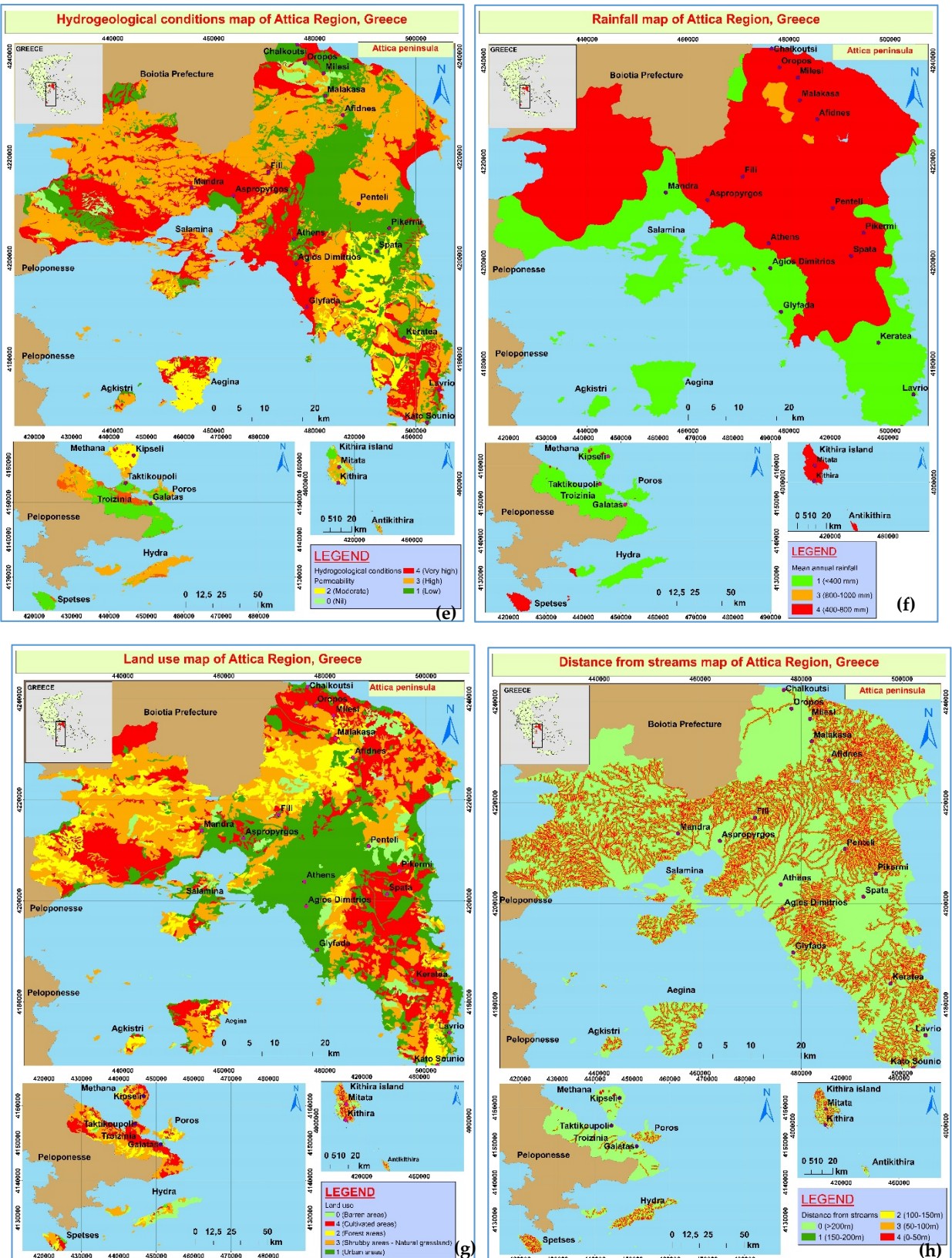

**Figure 10.** *Cont.*

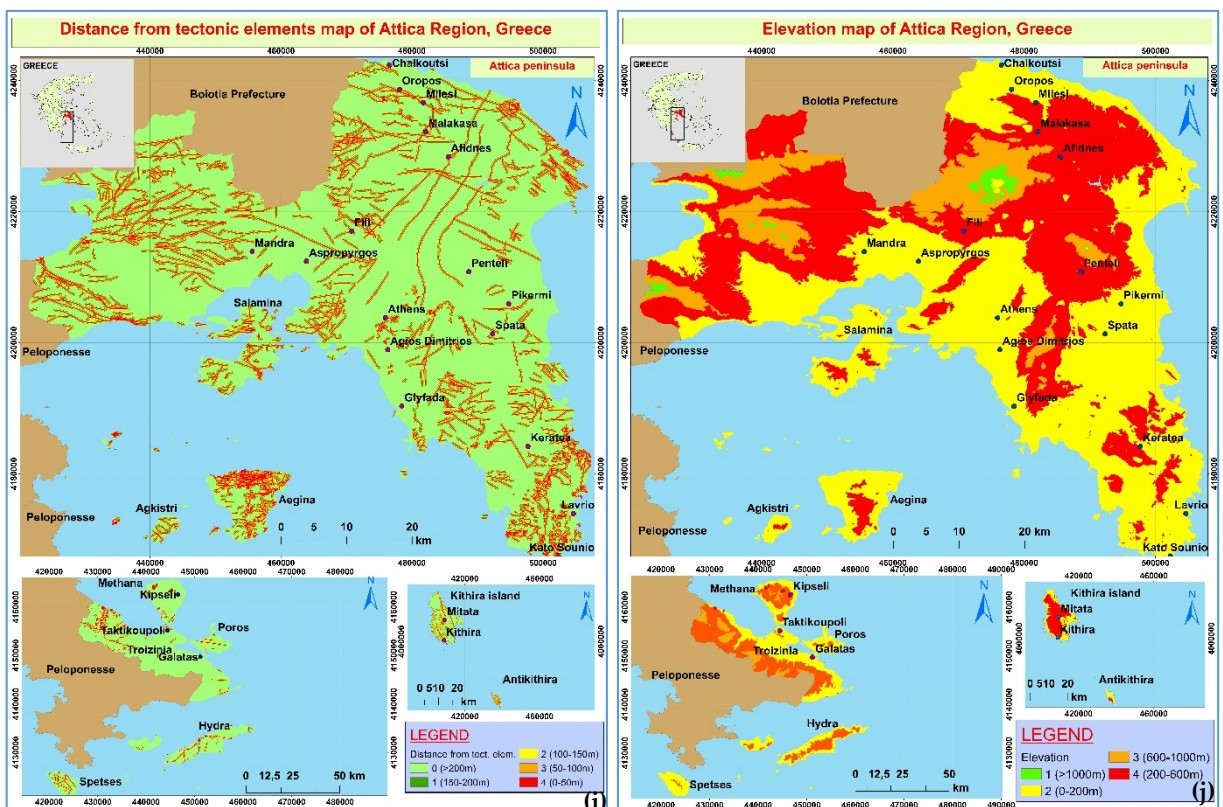

**Figure 10.** Thematic raster maps of the ten (10) landslide parameters used for the estimation of Attica region susceptibility: (**a**) Distance from roads, (**b**) Slope, (**c**) Aspect, (**d**) Reclassified geological map, (**e**) Hydrogeological conditions, (**f**) Rainfall, (**g**) Land use, (**h**) Distance from streams, (**i**) Distance from tectonic elements, (**j**) Elevation.

From Figure 11, some further findings that come out are as follows (Table 6, Figure 12):

**Table 6.** Correlation between instability index and susceptibility coverage class in km$^2$.

| Instability Index Category | Susceptibility Coverage Class in km$^2$ |
|---|---|
| <25% | 5 (0.13%) |
| 25.01–42% | 585 (15.54%) |
| 42.01–53% | 1552 (41.23%) |
| 53.01–70% | 1500 (39.85%) |
| 70.01–100% | 122 (3.24%) |
| | Total examined area: 3.764 km$^2$ |

From the above pie diagram, it is clear that 43.09% (39.85% + 3.24%) of the examined area is associated with an instability index greater than 53%. Furthermore, it can be added that 122 km$^2$ (3.24%) of the total examined area are correlated to potential landslide occurrence. Public authorities responsible for auditing and supervising technical works should be aware of these findings, so as to take the appropriate advance, mitigation measures against the possible initiation of potential disastrous landslide phenomena taken place in these proposed, for slope failures, areas.

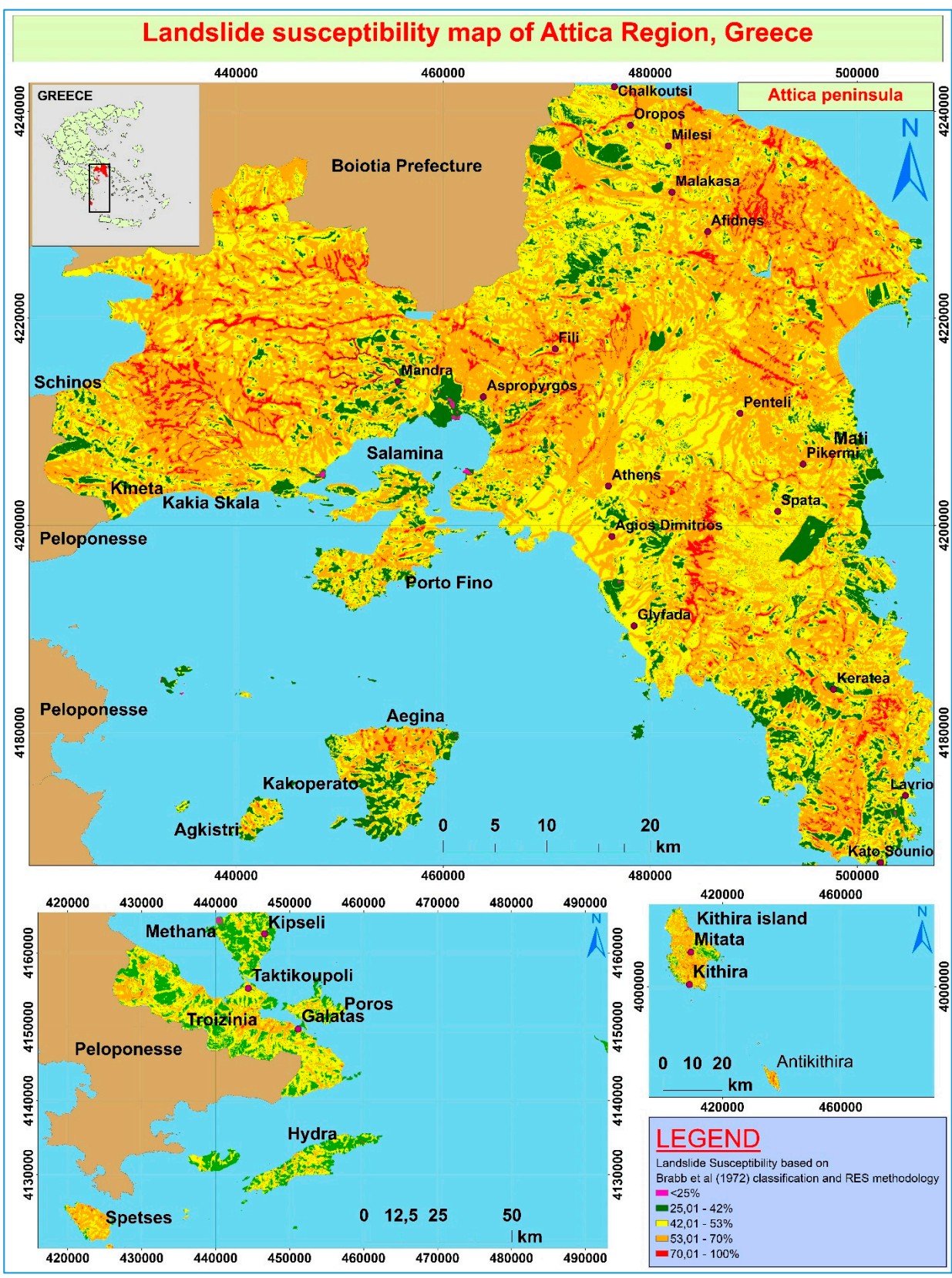

**Figure 11.** The Susceptibility map of Attica Region.

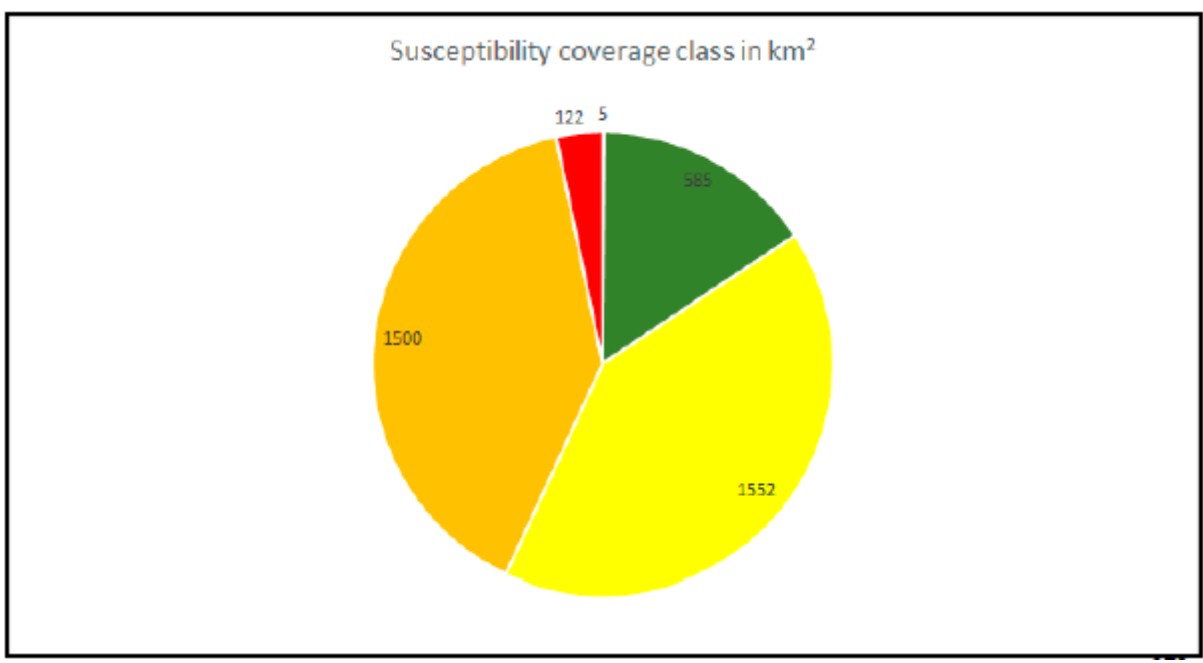

**Figure 12.** Pie diagram depicting the landslide susceptibility coverage class in km$^2$ for Attica region.

### 4.4. Validation of the Landslide Susceptibility Map

For having scientific significance in any generated model, the most important component in prediction modelling, is to implement a validation of the prediction results [51]. Thus, in the final landslide susceptibility map, we compared the results with the distribution of the 220 slope failure events that had occurred in the examined area. The predicted map showed very satisfactory results and particularly, at the susceptibility map of the Attica region, 68% of the locations of actual and potential landslides correspond to the "Extremely high" and 21% are associated with a landslide (Figure 13, Table 7).

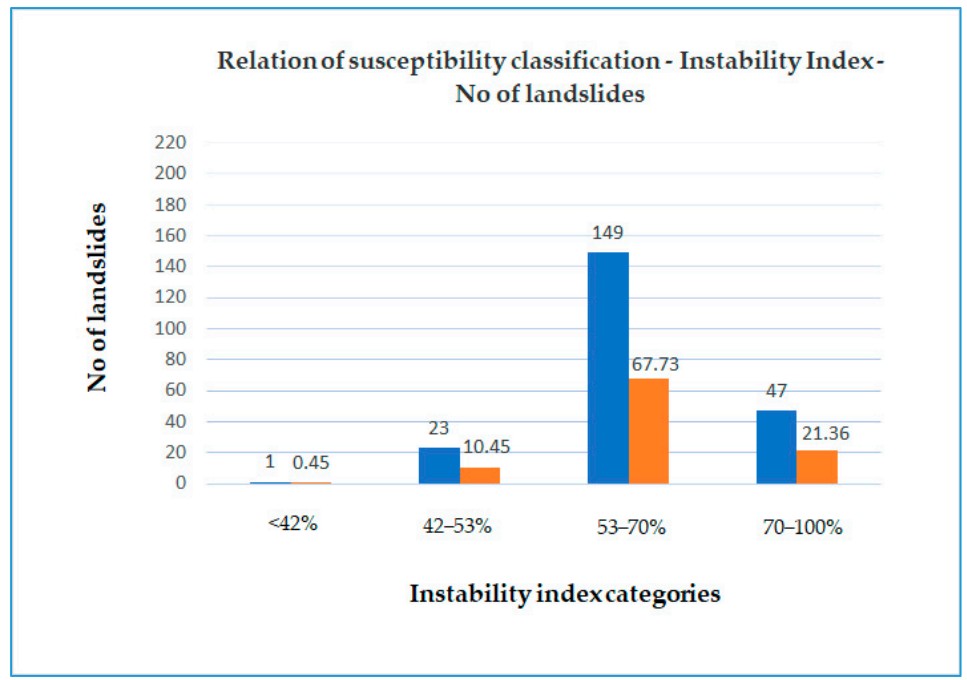

**Figure 13.** Correlation among number of examined slope failures, instability index, and susceptibility classification. Blue color corresponds to the number of slope failures, orange color is linked with instability index percentage associated with susceptibility categories.

**Table 7.** Correlation among number of examined slope failures, instability index, and susceptibility classification.

| Relative Susceptibility Classification | Number (No) of Slope Failures | Instability Index (%) |
|---|---|---|
| <42% (Low to moderate) | 1 | 0.45 |
| 42.01–53% (High) | 23 | 10.45 |
| 53.01–70% (Extrem. high) | 149 | 67.73 |
| 70.01–100% (Landslide) | 47 | 21.36 |
| | Total: 220 | 100 |

Moreover, another method for validating the above mentioned, was the implementation of the confusion matrix. It is a table that is often used to describe the performance of a classification model on a set of test data for which the true values are known [52]. We used the confusion matrix for a binary classifier e.g., ($\alpha$) the existence of landslides with instability index greater than 53% and (b) the no existence of landslides (with instability index less than 53%). Each row of the matrix represents the instances in an actual class while each column represents the instances in a predicted class (or vice versa). In our case, In Table 8, four different combinations of predicted and actual values were used.

**Table 8.** Confusion matrix of the landslide susceptibility map validation.

| Total Population: n = 220 Slope Failures | | Predicted Conditions | |
|---|---|---|---|
| | | Predicted NO | Predicted YES |
| True conditions (Observed) | Actual NO | 3 (TN) | 21 (FP) |
| | Actual YES | 22 (FN) | 174 (TP) |

Where: TN means when the examined slope does not correspond to landslide, how often does it predict no, FP means when the examined slope does not correspond to landslide, how often does it predict yes, FN means the falsely predicted landslide, TP means when the examined slope correspond to landslide, how often does it predict yes.

The following is a list of rates that were computed from the confusion matrix for a binary classifier:

- Accuracy: Overall, how often is the classifier correct?

$$(TP + TN)/total = (174 + 3)/220 = 0.80 \tag{4}$$

- Precision: Out of all the positive classes we predicted correctly, how many are actually positive.

$$TP/predicted\ yes = 174/195 = 0.89 \tag{5}$$

- Prevalence: How often does the yes condition actually occur in our sample?

$$actual\ yes/total = 196/220 = 0.89 \tag{6}$$

From the above, analytically presented, it is clear that the described RES methodology has 89% precision.

In addition, the validation of the generated susceptibility map was tested with two additional landslide databases. These are (a) the 98 polygons derived from Oregon methodology as previously mentioned, and (b) erosion lines derived from a project delivered by the Hellenic Survey of Geological and Mineral Exploration concerning the Mandra area flooding susceptibility [53]. Particularly, it was found that regarding the Oregon protocol, in the generated landslide susceptibility map of the Attica region, 49% of the

defined polygons correspond to the "Extremely high" category and 33% are associated with landslides. Concerning the rest of the delineated areas, it is proposed to conduct geological–geotechnical investigations to define the potential of the slopes to failure.

Finally, the erosion lines which were defined by the aforementioned research institute were in accordance with the instability index greater than 70%.

Practical use of the final susceptibility map is the implementation it may have during the planning, design and construction of various important infrastructure projects. Even though it is not advisable to be used for local or site-specific planning, C.J. Van Westen (2016) [54], recommends the following use of the above-mentioned susceptibility classes.

**Low susceptibility zones**

In those areas, with respect to planning and constructing civil engineering projects, no special care should be taken by planners and engineers.

**Moderate susceptibility zone**

This zone is the most problematic for spatial planning and construction infrastructure and it is encouraged to implement geotechnical/geophysical investigation for critical civil engineering projects (e.g., highways, important public buildings such as hospitals).

**High and very high susceptibility zones**

Slope failures are expected to occur within these zones. The best is to avoid these areas regarding the development of future residential areas or crucial infrastructure projects. However, if this is not possible, a detailed level of geotechnical investigation of landslide hazard is required for these areas before allowing new constructions. In the present study, areas of this category can be found in the northeastern, southeastern and western part of the Attica peninsula as well as in the northern part of Aegina island, and the central part of Salamina and Kithira islands.

## 5. Discussion

Using RES and GIS techniques, the landslide susceptibility of the Attica Region was assessed by correlating ten parameters and producing the final susceptibility map for the whole Attica peninsula (with its islands included) in Greece. The validity of this approach was tested using the slope failures that were recorded during the last sixty years in this region. In particular, 68% of the recorded 220 slope failures were found to be in the "Extremely high susceptibility" and 21% in the "Landslide" zone respectively of the developed map. Studying more carefully this map, some more remarks can be extracted.

Initially, it is shown in the susceptibility map that slope failure incidents are located mostly in areas where Neogene and Quaternary sediments outcrop. Secondarily, slope failures are associated with carbonate rocks basically due to rockfalls. In order to preliminary assess the potential landslide risk in respect to settlements, the villages and cities at the study area were plotted on the susceptibility map (Figure 11).

This correlation suggests that 16 settlements are entirely located within "Landslide" and 201 urban areas are in the "Extremely high landslide susceptibility" zone. To be more specific, in the "Landslide" zone, places such as Chalkoutsi, Grammatiko, Kato Alepochori, Schinos can be found. In the "Extremely high landslide susceptibility" zone, characteristic sites are Mesagros (Aegina Island), Varnavas, Galaniana (Antikithira island).

In addition, many defined slope failure areas are associated with the existence of faults. This result should be taken into consideration by public authorities responsible for the construction of public technical works, regarding urban planning and design of new infrastructure projects (e.g., highways, tunnels, major buildings).

According to the generated susceptibility map, areas associated with an instability index greater than 70% are located in many sites around the Attica region (islands included). For that reason, public authorities responsible for civil protection need to get advice from

such maps to make emergency plans at different administrative levels, useful for the pre-event of the landslide risk management cycle.

Moreover, the landslide susceptibility map can be used with the already produced potential highly flood hazard zoning maps of Attica Region authorized by the Greek Ministry of Environment and Energy, and with the produced flooded area maps, delivered by the Copernicus Emergency Management Service-Mapping.

Concerning localities that were affected by catastrophic forest fires in previous years such as those of Kineta (2018) and Mati (2018) (Figure 11) [38], and studying the generated susceptibility map, it is realized that such fires can cause in the immediate future "secondary" hazards like earth slides, debris flows and flash floods. Those two areas are associated with an instability index greater than 53% and this means that the drastically changed environmental conditions due to the fires may increase the landslide activity in the area in the near future.

Finally, it should be pointed out that the developed susceptibility map is at a regional scale (1:100,000) and its practical use is to be applied in conjunction with site-specific work, from experts such as experienced geologists, geotechnical engineers before development takes place. Additionally, it should be mentioned that even though susceptibility analysis does not define either the time and the type of the failure, or the volume of the mass involved, it is necessary for the estimation of hazard and risk index and zoning, respectively.

For all these reasons, the applied methodology (RES and Oregon Protocol methodology) should be accompanied each time by the appropriate fieldwork as well as the necessary geotechnical desk study, so as to acquire the most accurate geological model of the ad hoc examined area susceptible to slope failure [55].

## 6. Conclusions

This study presents the landslide susceptibility analysis for Attica Region, which is the most densely populated area in Greece. The produced susceptibility map is a cartographic product in a regional scale (1:100,000) generated for the Attica county via a semi-quantitative heuristic methodology named Rock Engineering System and a prototype technique originally developed by the Oregon Department of Geology (USA). To the author's knowledge, this is the first time that such an in-depth analysis has been conducted for the whole of Attica county. Furthermore, for the compilation of this map, RES methodology was applied as a simple and fast tool for the calculation of the instability index of each examined slope failure recorded in a well-organized geodatabase according to the EU Inspire Directive.

Considering the mentioned previously, it should be noted that 68% of the locations of actual and potential landslides correspond to the "Extremely high" and 21% are associated with a landslide. Responding to the previous remark, particular sites in Northeastern Attica (e.g., Kapandriti, Varnavas, Oropos, Kalamos), historical slopes in Western Attica (such as those of Alepochori-Psatha, Alepochori-Schinos, Kakia Skala), the most well-known historical landslide of Malakasa, characteristic places in Attica islands (e.g., Kithira-Kapsali, Aegina-Kakoperato, Salamina-Porto Fino) were validated through the above–mentioned methodology and it was found that all of them were confirmed as landslides (Figure 11). Furthermore, this correlation suggests that 16 settlements are entirely located within "Landslide" and 201 urban areas are in the "Extremely high landslide susceptibility" zone.

As in Section 3.2 is mentioned, RES methodology was applied in different physiographic environments with a variety of geological and tectonic settings and scales. In the present study, the previous statement was confirmed by implementing RES in an area with complex geological settings (e.g., active faults, many different streams based on Strahler classification as well as a variation of geological formations). Thus, it is suggested that this procedure (i.e., RES, GIS techniques, Oregon protocol-Special Paper 42) could be used in other regions with different geological environments and tectonic characteristics.

Summarizing, the DIAS geodatabase represents the spatial distribution of over 300 landslides (rockfalls, falls, erosion lines included) based on published and unpublished informa-

tion, field observations and remote sensing techniques. The intention is that the database should be updated constantly. The outcome of the DIAS project will be accessible to the public, through a web-based platform using an open-source G.I.S. software so as to aid awareness of landslides among different stakeholders (e.g., landslide experts, government agencies, planners, citizens). Moreover, the DIAS project can facilitate the role of Civil Protection Authorities, by providing inputs for prevention and preparedness.

Taking into consideration the previous outcomes, the upcoming steps of this research (DIAS project) will be the generation of hazard and risk maps using triggering dynamic factors like earthquake and rainfall data, as well as different elements of risk, respectively, in specific areas.

**Author Contributions:** Conceptualization, N.T. and G.P.; methodology, N.T. and G.P.; software, N.T. and P.A.; validation, N.T., G.P. and A.G.; formal analysis, N.T., G.P. and A.G.; investigation, N.T.; resources, N.T., G.P. and A.G.; data curation, N.T., G.P., A.G. and P.A.; writing—original draft preparation, N.T.; writing—review and editing, N.T., G.P., A.G. and P.A.; visualization, N.T.; supervision, G.P., A.G.; project administration, N.T., G.P.; funding acquisition, N.T., G.P., A.G. and P.A. All authors have read and agreed to the published version of the manuscript.

**Funding:** This research was co-funded by Greece and the European Union (European Social Fund-ESF) through the Operational Program "Human Resources Development, Education and Lifelong Learning 2014–2020" in the context of the project "Landslide Risk Assessment of Attica Region" MIS (5050327).

**Institutional Review Board Statement:** Not applicable.

**Informed Consent Statement:** Not applicable.

**Acknowledgments:** The authors are grateful to the Greek Ministry of Environment, Region of Attica, Hellenic Survey for Geology and Mineral Exploration (H.S.G.M.E.), General Secretary of Civil Protection of Greece and Greek Cadastre S.A. for providing valuable technical landslide reports as well as crucial digital geodata records. Furthermore, rainfall data were provided by the Institute for Environmental and Sustainable Development Research (IEPBA) of the National Observatory of Athens.

**Conflicts of Interest:** The authors declare no conflict of interest. The funders had no role in the design of the study; in the collection, analyses, or interpretation of data; in the writing of the manuscript, or in the decision to publish the results.

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
