# Peer review of "Development of the Landslide Susceptibility Map of Attica Region, Greece, Based on the Method of Rock Engineering System"

_land, doi:10.3390/land10020148_

Round 1
Reviewer 1 Report
Dear Authors,
your research is quality and the result (the landslide susceptibility map) is very useful. However, I have a few suggestions and comments for improvement that mostly relate to figures:
- Figures 1 and 2 - increase figures resolution (some details in the figures are less visible and blurry)
- Figures 3 and 9 - increase the visibility of graphic scales on individual maps that make up Figure 3
- Why "Scheme 1-4"? - in my opinion, these are the same Figures
- Figure 8 - in the PDF document I received for review parts of the figure (a-i) are improperly scattered and some parts are missing. The biggest problem is that the maps are too small and the visibility of the details is very poor.
Author Response
Response to Reviewer 1 Comments
Point 1: Figures 1 and 2 - increase figures resolution (some details in the figures are less visible and blurry)
Response 1: Changes have been made according to Reviewer’s first comment (Point 1).
Point 2: Figures 3 and 9 - increase the visibility of graphic scales on individual maps that make up Figure 3
Response 2: Changes have been made according to Reviewer’s second comment (Point 2).
Point 3: Why "Scheme 1-4"? - in my opinion, these are the same Figures
Response 3: Changes have been made according to Reviewer’s third comment (Point 3). Schemes 1-4 have been turned into Figures respectively.
Point 4: Figure 8 - in the PDF document I received for review parts of the figure (a-i) are improperly scattered and some parts are missing. The biggest problem is that the maps are too small and the visibility of the details is very poor.
Response 4: Changes have been made according to Reviewer’s fourth comment (Point 4).
Reviewer 2 Report
Estimating landslide susceptibility is very important for disaster risk management, so the theme of this paper is important. However, the aim of this study and the methodology of this study are not clearly explained in this manuscript. Therefore, the results of this research cannot be well understand in this manuscript.
- In the “1. Introduction”, the aim of this study is not clearly mentioned. Why Attica region is focused? The introduction of this manuscript seem to be just the introduction of survey report about Attica region, not academic research paper.
- It is mentioned that the method of this research “RES” was developed by Hudson and it has been applied by many researchers. The originality and new findings should be mentioned in this manuscript.
- In the “2. Geology and tectonic setting of Attica”, names of many regions are described, but the locations cannot be indicated in Figure 1 at all.
- The resolution of Figure 1 is too low to interpret the contents. Photos of figure 2 do not clearly indicate the situation of rock falls, especially, figure2c. Figures 8 are overlapped.
- The “RES” is not general method yet, so the basic concept of the “RES Approach” should be explained more in “3.2.2”
- In 3.2.2., it was mentioned that the ten parameters are independent. Is distance from roads is related to vegetation removal? Is hydrogeological condition related to lithology? It seems that the parameters are not independent.
- In table 1, the grade of parameters are shown, but the meaning of the thresholds are not mentioned. Author should explain how to decide the thresholds. Otherwise, the results cannot be understand well.
Author Response
Response to Reviewer 2 Comments
Estimating landslide susceptibility is very important for disaster risk management, so the theme of this paper is important. However, the aim of this study and the methodology of this study are not clearly explained in this manuscript. Therefore, the results of this research cannot be well understood in this manuscript.
Point 1: In the “1. Introduction”, the aim of this study is not clearly mentioned. Why Attica region is focused? The introduction of this manuscript seems to be just the introduction of survey report about Attica region, not academic research paper.
Response 1: Why Attica region is focused
This region which is a county of approximately 3800 km2, was selected because:
(i) in this region, many cases of slope failures have been reported; the well-known historical landslide of Malakasa (1995) which caused serious economic consequences due to the cut – off connection between Athens (the capital city of Greece) and the northern part of Greece; the dangerous, due to rockfalls, segment (located in Kakia Skala) of the National motorway connecting Athens to Patras, some other characteristic rockfall sites such as Alepochori – Psatha, and Alepochori – Schino in Western Attica, falls at particular seg-ments of main streams due to erosion and flash floods, landslides and rockfalls at Attica islands (e.g. Kithira, Salamina, Aigina, Spetses, Hydra, Poros), are some to name as the most characteristic slope failures already taken place in the administrative Region of Attica. Thus, since the principle that “slope failures in the future will be more likely to occur under the conditions which led to past and present instability” (Varnes, 1984.) can be adopted, inventorying and mapping the susceptible to failure slopes, is a crucial information for evaluating the future occurrence of landslides in this region.
(ii) the existing information considering the landslide occurrences in Attica Region was dispersed in more than one public agency, and was mainly focused on landslides documented along the road network and residential areas, while only few cases were georeferenced. The slope failures induced at the mountainous areas and at sites that are not close to the manmade environment were either not recorded or probably under-reported. Thus, there is a need for gathering every slope failure happened till nowadays, for generating hazard maps in order to use them for civil protection actions.
(iii) Attica region concentrates almost half of the Greek population, more than 60% of the industrial production in Greece and high value properties and infrastructure. For this reason, mapping areas prone to slope failure, helps public authorities associated with public works, in taking mitigation measures against the increase of risk in potentially dangerous areas, leading to losses of life and investments in such a densely populated county.
(iv) of the completeness and quality of the available slope failures and thematic geodata.
(v) to authors knowledge, this is the first time that a landslide susceptibility analysis has been conducted in a regional scale (1:100.000), for the whole territory of Attica Region. Furthermore, the generated landslide susceptibility map will serve for many authorities related to public works, as dynamic map for the planning, design, and implementation of a long-term landslide reduction strategy as well as identifying the areas where more de-tailed investigations will be required for the planning of critical infrastructure.
(vi) taking into account that the next five to ten years, very important civil engineering projects are about to be constructed in Attica county (such as transports network elements: highways, railroads, metro-tunnels, hospitals, administrative buildings, security/emergency structures, residential buildings) the existence of a regional scale landslide susceptibility map could be a very useful tool for supporting decisions in order to prevent the location of high-value constructions in unsuitable locations.
Point 2: It is mentioned that the method of this research “RES” was developed by Hudson and it has been applied by many researchers. The originality and new findings should be mentioned in this manuscript.
Response 2: In the manuscript, concerning the originality of RES, it is mentioned that RES was first introduced by J. Hudson in 1992. Furthermore, regarding new findings, it can be said that:
- Rafiee et al (2018), have used fuzzy RES in order to apply system thinking-based techniques for assessment of the rock mass cavability in block caving mines.
- Wang et al (2018) have implemented RES to evaluate sandy soil liquefaction.
- Ferentinou and M. Fakir (2018) used RES in accordance with self-organising maps (e.g., artificial neural networks), so as to assess the stability performance of newly open pit slopes.
- Finally, M. Elmouttie and P. Dean (2020), used RES and a system theoretic process analysis in order to design the control system for the slope stability monitoring in an open cut mining.
Point 3: In the “2. Geology and tectonic setting of Attica”, names of many regions are described, but the locations cannot be indicated in Figure 1 at all.
Response 3: The names of the regions that are mentioned in the “2 Geology and tectonic setting of Attica”, have been added in Figure 1.
Point 4: The resolution of Figure 1 is too low to interpret the contents. Photos of figure 2 do not clearly indicate the situation of rock falls, especially, figure2c. Figures 8 are overlapped.
Response 4: Changes have been made according to Reviewer’s fourth comment (Point 4).
Point 5: The “RES” is not general method yet, so the basic concept of the “RES Approach” should be explained more in “3.2.2”
Response 5: It has been added the following passage as an introduction to section 3.2.1:
A crucial problem of any engineering design is ensuring that all the necessary parameters have been included and that the interactions between them are understood. John Hudson was the researcher that originally introduced the Rock Engineering Systems (RES) approach in 1992. The RES methodology is a synthetic approach which studies the problem (e.g., landslide), breaks it down into its constituent variables (e.g., predisposing parameters, estimation of landslide instability index), and assesses their significance (e.g., calculation of susceptibility analysis). In most slopes, that kind of analysis is complicated due to different interacting factors, complexity of geological formations, different scale of the instability events as well as scarcity of detailed geodata. These problems can be solved through the use of RES, where its use can take into account the particular problems at any investigated site so as to identify critical sites in order to support decisions on land use and planning development.
For consideration of a specific engineering project – system (in our research the landslide susceptibility of Attica region), some parameters will have a greater effect on the project - system than others and some parameters will in their turn be significantly affected by the system. The RES methodology uses a table (i.e., interaction matrix) with xi rows and yj columns, in which the selected n parameters are selected as leading diagonal terms and the interactions between them are considered as off-diagonal terms.
Point 6: In 3.2.2., it was mentioned that the ten parameters are independent. Is distance from roads is related to vegetation removal? Is hydrogeological condition related to lithology? It seems that the parameters are not independent.
Response 6: Τhe meanings of “dependance” variable and “independent” parameter have to do with the role each one has inside the whole system we study. The system can be a slope failure or an underground stability and support or the selection of the right type of tunnel boring machine or any other geotechnical engineering problem that can be addressed by using this semi-quantitative heuristic methodology of RES.
To be more specific, by referring to “dependance” variable, it is meant the occurrence or not of a slope failure. For example, we study the interaction of ten landslide parameters and according to RES methodology we calculate the weighted coefficient of each landslide parameter, estimating the instability index for each examined slope. If the calculated index is over a critical accepted threshold (as the one we will present in the following section of this paper), this means the selected parameters are crucial for the slope failure occurrence, and subsequently measures must be taken in order to minimize their effect on slope instability. Otherwise, if the estimated index is under this critical threshold, then, no potential landslide is about to happen and immediately we conclude that those parameters that we have selected are not crucial for landslide initiation.
On the other side, “independent variables” are the landslide controlling factors (such as geology, distance from roads, hydrogeological conditions, distance from tectonic elements, etc.), which each other is tested on how dominant or how interactive can be with the other selected landslide parameters.
RES studies the interaction of each parameter to the other and vice versa, by quantifying the different importance of these interactions. This is justified, because some parameters will have a greater effect on the system (e.g., in our case the landslide susceptibility in Attica county,) than others and some parameters will in their turn be largely affected by the system. So, talking for example, about the interaction of hydrogeological condition on lithology, it is meant how lithology can be affected by the permeability status that dictates the geological formation that constitute the examined slope and vice versa how a specific type of rock or soil of the examined slope will affect the hydrogeological equilibrium of the slope. In another case, we examine how the distance from a road affects the amount of vegetation exists around this. To be more specific, if a public authority plans to construct a new highway in a place where forest or a grassland area already exists in that particular zone, then it has been proved that buffer zones of highway that are in a distance 50 or 200m from the surrounded slopes affect the existence of vegetation dramatically (Rozos et al., 2011). Vice versa, the influence of vegetation on slopes that are in a x distance from roads is less important.
Point 7: In table 1, the grade of parameters is shown, but the meaning of the thresholds is not mentioned. Author should explain how to decide the thresholds. Otherwise, the results cannot be understood well.
Response 7: For each parameter, an added explanation regarding its threshold has been written in the manuscript (Section 3.2.2).
Reviewer 3 Report
The Authors show the application of RES methodology for large-scale landslide susceptibility assessment in the Attica Region (Greece). The topic treated is very interesting and addressed with logical and scientific rigor. The results are also validated through the application of a simple statistical model.
However, some clarifications are necessary. I would recommend to introduce a specific chapter for the results, in the current form it is not present, but integrated in the chapter Materials and Methods.
Another important aspect are the Figures. Almost all of them are not very readable, in particular those perhaps the most important, 1 but especially 8. In almost all the figures toponyms, that could improve the reading, are missing.
Another aspect that the authors should improve on is the nomenclature of landslide types. Very often authors speak of landslides in general. It might be useful, and more scientific, to use the official classifications, such as those of Varnes 1978, Cruden and Varnes 1996, Hungr et al., 2014. In addition, a specific section could be introduced in the framework to illustrate the landslide types present in the study area. The example shown in Figure 2 is only about fall phenomena, are these the only phenomena? Finally, I would speak of predisposing parameters rather than "causual", "causative".....
Therefore, although it is an absolutely inherent theme of the journal, I believe it can only bepublished after careful review.
The following are some observations:
Lines 42-45: I would also talk more precisely about statistical methods
Line 172: the word Figure is missing the 2
Figure 2: make it bigger
Line 195: here they are talking about the time interval 1960-2020. In other parts of the text they indicate 1961-2020.
Scheme 1 and 2: what is the meaning? In the caption of Scheme 1 we talk about frequency. Frequency implies the concept of time, but no time intervals are represented in the figure.
Figure 3: Explain the meaning of existing and potential landslides. Give each figure a different label i.e. a); b)... Improve quality i.e. metric scale.
Line 320: I would not say from 1 to 10 but rather from 1 to n.
Line 404: I would talk about Land Use rather than Vegetation.
Section 3.2.3: should be put in the Results section.
Table 3: it is not very clear to me how to get it. It could be explained better
Figure 7: some labels of the axes are misspelled
The subdivision of the predisposing parameters into subclasses seems not to be used for the evaluation of the final Susceptibility. Correct?
It is not clear how to obtain the final susceptibility map. It might be useful to introduce a formula to make the concept more explicit.
Scheme 3 is a Figure. Check Km2
A review of English is necessary
Author Response
Response to Reviewer 3 Comments
The Authors show the application of RES methodology for large-scale landslide susceptibility assessment in the Attica Region (Greece). The topic treated is very interesting and addressed with logical and scientific rigor. The results are also validated through the application of a simple statistical model. However, some clarifications are necessary.
Point 1: I would recommend to introduce a specific chapter for the results, in the current form it is not present, but integrated in the chapter Materials and Methods.
Response 1: Changes have been made according to Reviewer’s first comment (Point 1). A new section (4. Results) has been added in the manuscript.
Point 2: Another important aspect are the Figures. Almost all of them are not very readable, in particular those perhaps the most important, 1 but especially 8. In almost all the figures toponyms, that could improve the reading, are missing.
Response 2: Changes have been made according to Reviewer’s second comment (Point 2). Resolution has been improved and toponyms have been added in the maps.
Point 3: Another aspect that the authors should improve on is the nomenclature of landslide types. Very often authors speak of landslides in general. It might be useful, and more scientific, to use the official classifications, such as those of Varnes 1978, Cruden and Varnes 1996, Hungr et al., 2014.
Response 3: Changes have been made according to Varnes classification (1978).
Point 4: In addition, a specific section could be introduced in the framework to illustrate the landslide types present in the study area. The example shown in Figure 2 is only about fall phenomena, are these the only phenomena?
Response 4: Changes have been made according to Reviewer’s forth comment (Point 4). In addition, pictures from another characteristic slope failures have been added in order to emphasize the variety of the slope failure types that have occurred in Attica county (e.g., complex slope failure taken place at Porto Fino site in Salamina island or earth slide in Penteli area, north of Athens).
Point 5: Finally, I would speak of predisposing parameters rather than "causual","causative"…..
Response 5: The proposed expression has been inserted in the manuscript.
Point 6: Lines 42-45: I would also talk more precisely about statistical methods
Response 6: Here is the added paragraph, as an answer to Point 6:
Internationally, three main data – driven approaches are mostly used: bivariate statis-tical analysis, multivariate statistical models and data integration methods like Artificial Neural Network analysis. Bivariate statistical methods (e.g., fuzzy logic, Bayesian combination rules, weights of evidence modelling) are an important tool that can be used in order to analyze which factors play a significant role in slope failure, without taking into account the interdependence of parameters. Multivariate statistical models evaluate the combined relationship between the slope failure and a series of landslide controlling factors. In this type of analysis, all relevant landslide parameters are sampled either on a grid basis or in slope unit and the presence or absence of landslides is evaluated. These techniques have become standard in regional scale landslide susceptibility assessment.
Point 7: Line 172: the word Figure is missing the 2
Response 7: Changes have been made according to Reviewer’s seventh comment (Point 7).
Point 8: Figure 2: make it bigger
Response 8: Changes have been made according to Reviewer’s eighth comment (Point 8).
Point 9: Line 195: here they are talking about the time interval 1960-2020.In other parts of the text they indicate 1961-2020.
Response 9: We have addressed this comment. The right chronological period is from 1961 up to 2020.
Point 10: Scheme 1 and 2: what is the meaning? In the caption of Scheme 1 we talk about frequency. Frequency implies the concept of time, but no time intervals are represented in the figure.
Response 10: In Scheme 1 (in the revised manuscript it has been turned into Figure 3), we refer to the relation between the number of landslides occurred per decade since 1961. This is expressed by the term “bin-size ten years”. However, we have fixed the title of y axis by writing “number of landslides per decade” instead of “frequency” and we have added in x axis intervals such as 1961-1970, 1970-1980, instead of 1960, 1970, etc.
Regarding the remark about Scheme 2 (which in the revised manuscript has been turned into Figure 7), it is mentioned that the sum of cause-and-effect value for each parameter represents how active that parameter is within the matrix system (i.e., the slope stability). Thus, if for example the landslide parameter “hydrogeological conditions” has the greatest value (concerning C+E), this explains that those conditions play the most decisive role for landslide activation.
Point 11: Figure 3: Explain the meaning of existing and potential landslides. Give each figure a different label i.e., a); b). Improve quality i.e. metric scale.
Response 11: We took into account this comment, by reconsidering these two terms “existing” and “potential”. Thus, we grouped – simplified the 220 slopes inventoried (as points in green circle), depicting them as one layer (e.g., slope failures_DIAS). Furthermore, we correspond the 98 slope failures delineated as polygons through Oregon methodology to the meaning of “potential” for the facility of the reader. In addition, metric’s scale quality has been improved.
Point 12: Line 320: I would not say from 1 to 10 but rather from 1 to n.
Response 12: For our case study, the implementation of RES is referred to ten landslide parameters. However, in general, indeed, the appropriate number corresponds to “n” slope failure parameters.
Point 13: Line 404: I would talk about Land Use rather than Vegetation.
Response 13: The authors agree with this remark. Vegetation has been changed into land use.
Point 14: Section 3.2.3: should be put in the Results section.
Response 14: The authors agree with this remark. A new section (4. Results, 4.1 Implementation of RES for the estimation of weighted coefficients) has been added.
Point 15: Table 3: it is not very clear to me how to get it. It could be explained better
Response 15: In this table, each examined slope (is depicted in the column “Slopes”) is ranked according to Table 2 (in the revised manuscript) rating, taking into account in parallel the specific geological conditions that characterize it according to either the ad-hoc technical report we have collected or field study we have done. Afterwards, for each slope site, every ranking of each parameter (each parameter is depicted in the second line under the title “Parameters”, named as 1, 2, 3, …, 10) is multiplied by its weighted coefficient (last line of the Table) respectively and each outcome, based on equation (1) is added in order to yield the instability index for each slope. For example, the instability index of Slope (1) is estimated as follows:
Σ [Parameter (1): 4 * 2,39 + Parameter (2): 1 * 2,77 + …… + Parameter (10): 2 * 1,72]=71.
Point 16: Figure 7: some labels of the axes are misspelled
Response 16: Changes have been made according to Reviewer’s comment (Point 16).
Point 17: The subdivision of the predisposing parameters into subclasses seems not to be used for the evaluation of the final Susceptibility. Correct?
Response 17: In the section “Landslide Susceptibility map”, the following passage has been written (we have included additional ad-hoc remarks relating to the Point 17):
“Afterwards, weights (from RES) and rank values to the reclassified raster layers (representing landslide factors) [this has been achieved through GIS algorithms] and to the classes of each layer (from Table 1) were assigned, respectively. This was realized with the use of the previously extended analyzed methodology of RES. Finally, the weighted raster thematic maps with the assigned ranking values for their classes (from Table 1) were multiplied by the corresponding weights (via RES) and added up (through the ArcGIS tool of weighted sum) to yield the slope failure map where each cell has a certain landslide susceptibility index value. The reclassification of this map represents the final susceptibility map of the study area, divided into susceptibility zones according to Brabb et al (1972) [53] classification (Figure 9)”. As a conclusion, we are saying that the subdivision of the predisposing parameters into subclasses, indeed, get used for the evaluation of the final susceptibility.
Point 18: It is not clear how to obtain the final susceptibility map. It might be useful to introduce a formula to make the concept more explicit.
Response 18: In the section “Landslide Susceptibility map”, it is written:
“the weighted raster thematic maps with the assigned ranking values for their classes were multiplied by the corresponding weights and added up (through the ArcGIS algorith of weighted sum) to yield the slope failure map where each cell has a certain landslide susceptibility index value. The reclassification of this map represents the final susceptibility map of the study area, divided into susceptibility zones according to Brabb et al (1972) [53] classification (Figure 9)”.
So, the formula the Reviewer mentions is included in the ArcGIS software (it is called Weighted Sum) and what it does, is adding the product of each layer (each selected landslide parameter) times its weighted coefficient respectively. Mathematically, we use the equations (1), (2) and (3).
Point 19: Scheme 3 is a Figure. Check Km2.
Response 19: Changes have been made according to Reviewer’s comment (Point 19).
Round 2
Reviewer 2 Report
The authors improve their document by answering the question and following the comments.
However, it need to be improve in a few points.
1.
Authors expressed the term GIS by "G.I.S" and "GIS".
It need to be unified.
2. The result of the research should be widely useful in other areas. Please also describe whether result of this research can be applied in other areas.
Author Response
Response to Reviewer 2 Comments (round 2)
The authors improve their document by answering the question and following the comments. However, it needs to be improved in a few points.
Point 1: Authors expressed the term GIS by "G.I.S" and "GIS". It needs to be unified.
Response 1: Changes have been made according to Reviewer’s first comment (Point 1). We decided to use GIS term in the manuscript.
Point 2: The result of the research should be widely useful in other areas. Please also describe whether result of this research can be applied in other areas.
Response 2: As in Section 3.2 is mentioned, RES methodology has been applied in different physiographic environments with a variety of geological and tectonic settings and scales. This is clearly referred from line 328 to line 356.
In the present study, the previous statement was confirmed by implementing RES in an area with complex geological setting (e.g., active faults, many different streams based on Strahler classification as well as a variation of geological formations).
Thus, it is suggested (and added in Section 6. Conclusions) that this procedure (i.e., RES, GIS techniques, Oregon protocol – Special Paper 42) could be used in other regions with different geological environments and tectonic characteristics.